# Attenuated dopamine signaling after aversive learning is restored by ketamine to rescue escape actions

**Mingzheng Wu, Samuel Minkowicz, Vasin Dumrongprechachan, Pauline Hamilton, Lei Xiao, Yevgenia Kozorovitskiy***

Department of Neurobiology, Northwestern University, Evanston, United States

**Abstract** Escaping aversive stimuli is essential for complex organisms, but prolonged exposure to stress leads to maladaptive learning. Stress alters neuronal activity and neuromodulatory signaling in distributed networks, modifying behavior. Here, we describe changes in dopaminergic neuron activity and signaling following aversive learning in a learned helplessness paradigm in mice. A single dose of ketamine suffices to restore escape behavior after aversive learning. Dopaminergic neuron activity in the ventral tegmental area (VTA) systematically varies across learning, correlating with future sensitivity to ketamine treatment. Ketamine's effects are blocked by chemogenetic inhibition of dopamine signaling. Rather than directly altering the activity of dopaminergic neurons, ketamine appears to rescue dopamine dynamics through actions in the medial prefrontal cortex (mPFC). Chemogenetic activation of Drd1 receptor positive mPFC neurons mimics ketamine's effects on behavior. Together, our data link neuromodulatory dynamics in mPFC-VTA circuits, aversive learning, and the effects of ketamine.

*For correspondence: Yevgenia.Kozorovitskiy@ northwestern.edu

Competing interest: The authors declare that no competing interests exist.

## Introduction

Major depressive disorder (MDD) is a prevalent mental illness linked to diminished quality of life and increased mortality. Persistent changes in mood and emotional reactivity represent fundamental features of MDD, extensively investigated in human subjects (*Rottenberg, 2017*; *Rottenberg et al., 2005*). Reduced reactivity to both positive and negative stimuli has been consistently observed in clinically depressed patients (*Bylsma et al., 2008*; *Rottenberg and Hindash, 2015*), suggesting that MDD may involve systematic changes in the processing of reward and aversion. These changes in reward-based and aversive responses can be modeled in animals (*Abler et al., 2007*; *Heldt et al., 2007*; *Nestler and Carlezon, 2006*; *Proulx et al., 2014*). In animal models involving prolonged stress, the reactivity to positive valence (e.g. social stimuli) and negative valence (e.g. tail suspension) experiences is usually diminished (*Beyer and Freund, 2017*), suggesting that prolonged aversive experience induces maladaptive learning. One established model of aversive learning is learned helplessness (LH) (*Abramson et al., 1978*; *Maier and Seligman, 1976*; *Seligman and Maier, 1967*). Following prolonged inescapable stress exposure, animals learn that outcomes are independent of their behavioral actions; this learning eventually diminishes attempts to escape from avoidable stressful stimuli (*Maier and Seligman, 2016*). This form of aversive learning has been reproduced in humans and other animals, including rodents (*Abramson et al., 1978*; *Chourbaji et al., 2005*; *Maier, 1984*; *Maier and Seligman, 2016*). Reduced reactivity to aversive stimuli after LH is reversed by antidepressant treatments in animal models (*Belujon and Grace, 2014*; *Chourbaji et al., 2005*; *Krishnan and Nestler, 2011*). Several studies have implicated the involvement of neuromodulatory systems, including dopamine, norepinephrine, and serotonin in the acquisition of LH and its expression over time (*Belujon and Grace, 2017*; *Eley et al., 2004*; *Nestler and Carlezon, 2006*; *Nutt et al., 2006*).

**eLife digest** Over 264 million people around the world suffer from depression, according to the World Health Organization (WHO). Depression can be debilitating, and while anti-depressant drugs are available, they do not always work. A small molecule drug mainly used for anesthesia called ketamine has recently been shown to ameliorate depressive symptoms within hours, much faster than most anti-depressants. However, the molecular mechanisms behind this effect are still largely unknown.

Most anti-depressant drugs work by restoring the normal balance of dopamine and other chemical messengers in the brain. Dopamine is released by a specialized group of cells called dopaminergic neurons, and helps us make decisions by influencing a wide range of other cells in the brain. In a healthy brain, dopamine directs us to rewarding choices, while avoiding actions with negative outcomes. During depression, these dopamine signals are perturbed, resulting in reduced motivation and pleasure. But it remained unclear whether ketamine's anti-depressant activity also relied on dopamine.

To investigate this, Wu et al. used a behavioral study called "learned helplessness" which simulates depression by putting mice in unavoidable stressful situations. Over time the mice learn that their actions do not change the outcome and eventually stop trying to escape from unpleasant situations, even if they are avoidable. The experiment showed that dopaminergic neurons in an area of the brain that is an important part of the "reward and aversion" system became less sensitive to unpleasant stimuli following learned helplessness. When the mice received ketamine, these neurons recovered after a few hours.

Individual mice also responded differently to ketamine. The most 'resilient', stress-resistant mice, which had distinct patterns of dopamine signaling, also responded most strongly to the drug. Genetic and chemical manipulation of dopaminergic neurons confirmed that ketamine needed intact dopamine signals to work, and revealed that it acted indirectly on dopamine dynamics via another brain region called the medial prefrontal cortex.

These results shed new light on how a promising new anti-depressant works. In the future, they may also explain why drugs like ketamine work better for some people than others, ultimately helping clinicians select the most effective treatment for individual patients.

Dopaminergic (DA) neurons in the ventral tegmental area (VTA) primarily encode reward and aversion, responding to both types of stimuli (*Lammel et al., 2014*; *Morales and Margolis, 2017*; *Tan et al., 2012*; *Tsai et al., 2009*; *Watabe-Uchida et al., 2017*). The activation of DA neurons is important in the development of aversive conditioning (*de Jong et al., 2019*), and the activity of DA neurons is differentially modulated by acute and chronic stress (*Grace, 2016*; *Hollon et al., 2015*; *Russo and Nestler, 2013*). Establishing a causal connection between DA neuron activity and depressive-like behavior, optogenetic activation of VTA DA neurons increases behavioral resilience to social defeat stress (*Chaudhury et al., 2013*) and promotes active coping actions (*Tye et al., 2012*). How VTA DA neurons adjust their activity during aversive learning and how these activity changes relate to reduced reactivity to aversive stimuli after learning remains unclear.

Widely used antidepressants, mostly targeting monoamine reuptake systems, are limited by delayed onset of efficacy, incomplete remission, and low remission rates (*Walker et al., 2015*). Ketamine acts primarily as an antagonist at the glutamatergic N-methyl-D-aspartate (NMDA) receptors, showing rapid onset anti-depressant effects in depressed patients (*Berman et al., 2000*; *Daly et al., 2018*). In addition, mechanisms beyond direct NMDAR antagonism likely participate in the rapid behavioral effects of ketamine, implicating other classes of glutamate receptors, neuromodulators, and emergent circuit-level dynamics (*Ali et al., 2020*; *Chatterjee et al., 2012*; *Duman, 2018*; *Hare et al., 2020*; *Lorrain et al., 2003*; *Zanos et al., 2016*). Ketamine ameliorates depressive-like behaviors in animal models of acute and prolonged stress (*Duman et al., 2016*; *Duman and Aghajanian, 2012*; *Fuchikami et al., 2015*; *Harmer et al., 2017*; *Krishnan and Nestler, 2011*), and rescues escape actions in response to aversive stimuli after LH induction (*Belujon and Grace, 2014*). A recently published meta-analysis suggests that acute sub-anesthetic doses of ketamine may increase DA levels in the cortex, dorsal striatum, and nucleus accumbens (*Kokkinou et al., 2018*). In vivo recordings

from electrophysiologically identified VTA DA neurons in rats highlight ketamine's modulation of neuronal firing (*Belujon and Grace, 2014*). The medial prefrontal cortex (mPFC), essential for higher order cognitive functions including the control of emotional processing, is one key site for ketamine effects in the brain. Ketamine has been shown to modulate mPFC activity and plasticity to rescue depressive-like behaviors (*Li et al., 2010*; *Lorrain et al., 2003*; *Moda-Sava et al., 2019*; *Ng et al., 2018*; *Shirayama and Hashimoto, 2017*; *Wu et al., 2021a*). Given the bidirectional connectivity between VTA and mPFC (*Beier et al., 2015*), ketamine may regulate VTA DA activity through actions in mPFC, proposed to potentiate the activity of pyramidal neurons rapidly through disinhibition by suppressing inhibitory interneurons (*Ali et al., 2020*; *Hare et al., 2020*; *Homayoun and Moghaddam, 2007*). Despite some reported and suggested links between ketamine and DA systems, the causal relationship between the DA system and ketamine's effects on behavior remains to be elucidated.

Here, we use fiber photometry to record the responses of VTA DA neurons to aversive stimuli across different phases of learning in LH. By leveraging the tunability of LH induction parameters, we also design a modified, weaker LH paradigm to reveal the activity patterns of VTA DA neurons for distinct behavioral outcomes, correlating the activity signatures with future sensitivity to ketamine treatment. By using chemogenetic inhibition, we demonstrate that VTA DA activity and downstream signaling is necessary for the behavioral effects of ketamine. Finally, combining fiber photometry, anatomical tracing, and chemogenetics, we find that mPFC serves as an action site of ketamine to restore DA dynamics and escape actions.

## Results

### VTA DA neuron activity during aversive learning

To define the function of midbrain DA neurons during aversive learning, we used a variant of learned helplessness (LH) (*Chourbaji et al., 2005*). A shuttle box with two compartments connected by a door allows animals to escape from one side to the other when an electric foot shock is delivered to either compartment. Prior to LH induction, mice were exposed to 30 escapable shocks to test baseline escape behaviors. Initially, mice escape from electric foot shocks. However, following repeated exposure to *inescapable* foot shocks, mice reduce escapes from avoidable 10 s-long foot shocks (*Figure 1a and b*). Our data and prior publications (*Belujon and Grace, 2014*; *Beurel et al., 2011*; *Maeng et al., 2008*) show that a single low dose of racemic ketamine 4 hr prior to the test (10 mg/kg, b.w., i.p.) is sufficient to rescue escape behavior in this LH paradigm (*Figure 1b*). A separate group of mice that received saline instead of ketamine following the same experimental design did not decrease escape failures after LH (*Figure 1—figure supplement 1a*). Thus, reduced failures to escape after ketamine treatment are not simply a function of time elapsed since LH induction (i.e. spontaneous fear extinction). Reduced escape behavior after LH induction, as well as its reversal by ketamine, are not strictly context-dependent, since the proportion of failures to escape was similar regardless of whether behavioral evaluation was carried out in the induction context or in a novel environment (*Figure 1—figure supplement 1b*).

To understand the relationship between VTA DA neuron activity and ketamine's rescue of escape behavior, we used fiber photometry to monitor the activity of the genetically encoded calcium indicator GCaMP6f in VTA DA neurons. DAT[icre] neonates were virally transduced with Cre-dependent AAV1. CAG.FLEX.GCaMP6f. Four to six weeks after transduction, optical fibers were implanted in the VTA (*Figure 1c*), guided by real-time photometry (*Figure 1—figure supplement 2a*). During fiber implant procedure, we observed that surgical anesthesia was associated with ~1 Hz oscillations in VTA calcium transients (*Figure 1—figure supplement 2b*). After mice recovered from surgery, we evaluated the activity of VTA DA neurons during aversive learning in young adult mice (P40-60) of both sexes. We first recorded $Ca^{2+}$ transients of VTA DA neurons in response to brief, inescapable foot shocks (3 s, 0.3 mA) during the learning period and after ketamine treatment. At the start of learning (Induction start), the activity of VTA DA neurons first decreased during the aversive foot shocks and then rose after the termination of the shock (*Figure 1d and e*). This biphasic response is consistent with recently published observations of VTA DA activity during other forms of aversive conditioning (*de Jong et al., 2019*). Notably, the responses of VTA DA neurons to inescapable foot shocks were blunted at the end of the second-day induction (Induction end). A single low dose of racemic ketamine (10 mg/kg, b.w., i.p., LH + KET) largely restored the characteristic $Ca^{2+}$ transient features, in parallel to the behavioral

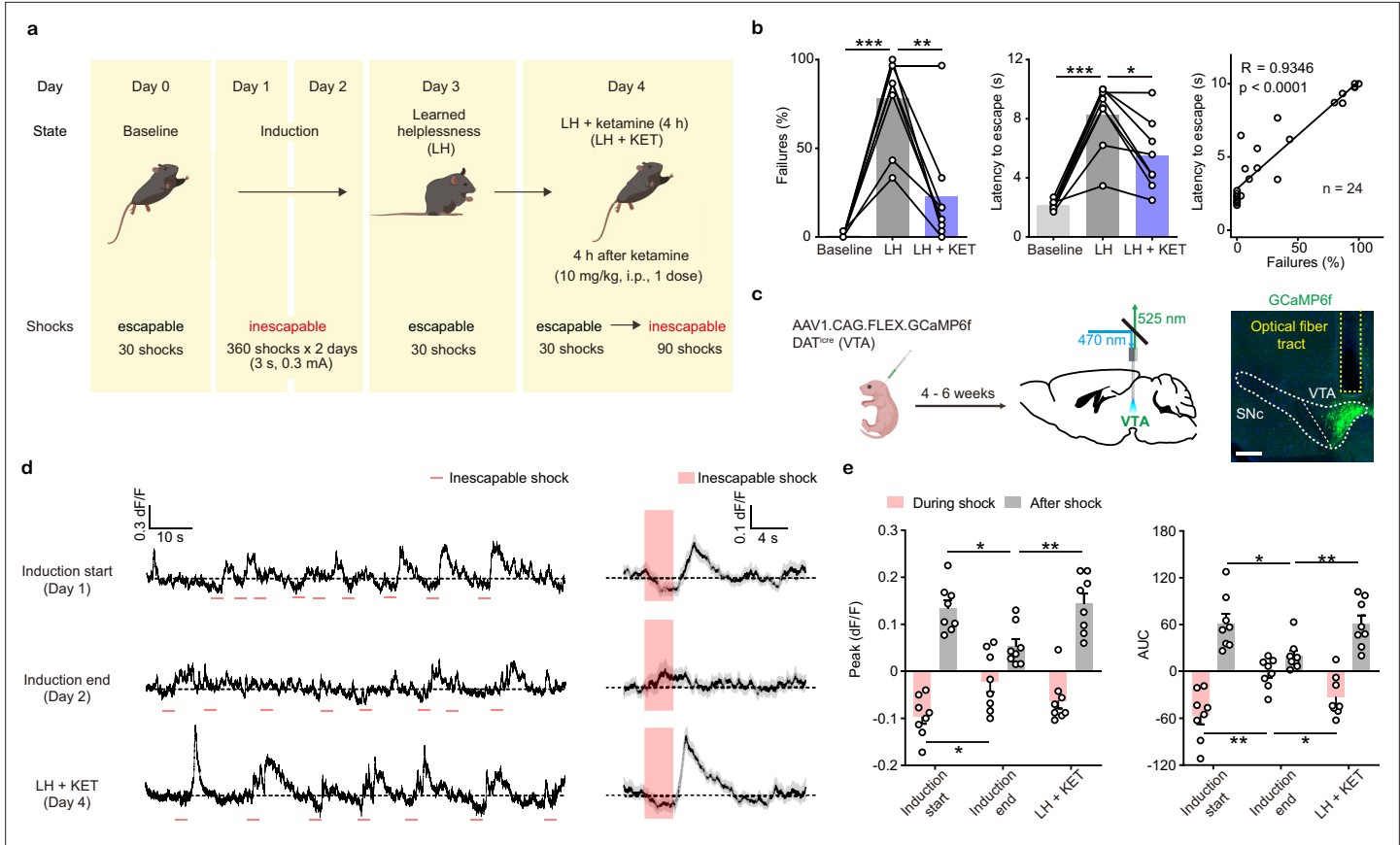

**Figure 1.** Ketamine rescues escape behavior and dampened DA neuronal activity after aversive learning. (**a**). Schematic illustrating the timeline of behavioral and pharmacological manipulations in the LH paradigm. (**b**). Left, summary data showing the percentage of failures to escape an escapable aversive shock across phases of learning (Baseline, LH, and LH+ KET). Middle, same as left, but for latency to escape. Right, for all conditions, the correlation between percentage of failure to escape and escape latency (failure to escape trials scored as 10 sec latency). n = 24 trials from eight mice. % Failures: repeated measures one-way ANOVA, F (1.89, 13.23) = 27.9, p = 0.0001, Sidak's multiple comparison test, Baseline vs LH, p = 0.0001, LH vs LH+ KET, p = 0.0037. Latency to escape: repeated measure one-way ANOVA, F (1.96, 13.72) = 29.63, p < 0.0001, Sidak's multiple comparison test, Baseline vs LH, p = 0.0003, LH vs LH+ KET, p = 0.0141. Pearson correlation: R = 0.9346, p < 0.0001. (**c**). Left, schematic for viral transduction in the VTA and subsequent fiber implant. Right, fiber placement verification. Green, GCaMP6f; blue, Hoechst nuclear stain. Scale bar: 500 μm. (**d**). Left, baseline adjusted raw traces of VTA DA neuron Ca²⁺ responses to inescapable foot shocks (3 s, pink) in one animal, at the start of induction, at the end of induction, and 4 hr following a single dose of ketamine (LH+ KET, 10 mg/kg i.p.). Right, average traces in the same subject aligned to shock start time (20 trials/condition, mean ± SEM). (**e**). Left, quantification of peak Ca²⁺ transient amplitude during and after foot-shock stimuli across conditions. Right, same but for area under the curve (AUC). Both positive and negative values are quantified. n = 8 animals, repeated measures one-way ANOVA, Holm-Sidak's multiple comparison test, Peak: During shock, F (1.823, 12.76) = 5.387, p = 0.0222, Induction start vs Induction end, p = 0.0458, Induction end vs LH + KET, p = 0.0788. After shock, F (1.693, 11.85) = 6.805, p = 0.0132, Induction start vs Induction end, p = 0.0230, Induction end vs LH + KET, p = 0.0068. AUC: During shock, F (1.705, 11.94) = 8.942, p = 0.0054, Induction start vs Induction end, p = 0.0058, Induction end vs LH + KET, p = 0.0258. After shock, F (1.437, 10.06) = 5.499, p = 0.0318, Induction start vs Induction end, p = 0.0257, Induction end vs LH + KET, p = 0.0069. *p < 0.05, **p < 0.01, ***p < 0.001. Error bars reflect SEM.

The online version of this article includes the following figure supplement(s) for figure 1:

**Source data 1.** Numerical data for the graphs in *Figure 1*.

**Figure supplement 1.** Characterization of behavior after learned helplessness.

**Figure supplement 1—source data 1.** Numerical data for the graphs in *Figure 1—figure supplement 1*.

**Figure supplement 2.** VTA DA neuron responses to anesthesia, foot shock, and motion transitions.

**Figure supplement 2—source data 1.** Numerical data for the graphs in *Figure 1—figure supplement 2*.

rescue (*Figure 1d and e*; *Figure 1—figure supplement 2d*, e). Visualizing sequential Ca²⁺ traces from individual trials illustrates (1) the prevalence of after-shock peaks at the start of learning, (2) their decreased latency relative to shock onset during the second day of training, and (3) the recovery of Ca²⁺ transient latency to peak following ketamine administration (*Figure 1—figure supplement 2f*). To quantify this temporal structure, we computed the latency to peak on sequential averaged trials (n = 10 trials/avg) and plotted the data as a scatter plot and cumulative distribution (*Figure 1—figure supplement 2g*). No significant transients were observed in GFP-expressing controls in this behavioral assay (*Figure 1—figure supplement 2c*).

Since the activity of DA neurons has been linked to movement (*da Silva et al., 2018*; *Howe and Dombeck, 2016*), one potential explanation for how aversive learning could modulate Ca²⁺ activity of VTA DA neurons involves altered locomotor behavior. While ketamine acutely changes locomotion, this effect normally occurs within tens of minutes following administration (*Ali et al., 2020*), and resolves by the time clinically relevant changes in affective behavior are observed (*Berman et al., 2000*; *Peltoniemi et al., 2016*). We found no differences in open-field locomotion across phases of learning (*Figure 1—figure supplement 2h*). Additionally, changes in VTA DA neuron activity were not associated with motion transitions, including onset and offset of locomotion (*Figure 1—figure supplement 2i*). This result is not surprising, since movement transitions are typically associated with the activity of DA neurons in the Substantia Nigra pars Compacta rather than the VTA (*da Silva et al., 2018*; *Howe and Dombeck, 2016*). Together, our data demonstrate that VTA DA neuron activity is restructured by LH, which is rescued following ketamine treatment.

## Outcome-specific VTA DA dynamics during weaker aversive learning and after ketamine treatment

To better understand the relationship between VTA DA neuron activity and specific behavioral responses across phases of learning, we designed a weaker learning paradigm (wLH), which allowed us to compare Ca²⁺ transients across distinct behavioral outcomes, escapes versus failures. wLH includes a larger number of brief escapable foot shocks as the test stimuli (3 s long, 100 trials), with only a single day of LH induction with inescapable shocks (*Figure 2a*). As anticipated, a weaker form of LH, characterized by a lower average failure rate, was observed in this paradigm compared to stronger LH (sLH, as in *Figure 1*) (two-way ANOVA for sLH and wLH across behavioral states, Sidak's multiple comparison, sLH 78 % vs wLH 48%, p = 0.0022). Again, escape behavior recovered after ketamine treatment, compared to saline-treated animals (*Figure 2b*, right panel). Another group of control animals underwent test sessions but no induction; they showed no changes in escape behavior over the time of the experiment (one-way ANOVA, p = 0.9882).

A three-state transition model depicts the probability of transitioning between behavioral responses, allowing us to compare patterns of response sequences across learning (*Figure 2—figure supplement 1a*). In addition to escapes and failures, responses were labeled 'spontaneous' when the animal spontaneously ran to the other side of the chamber during the random length pretrial time. Baseline and post-ketamine sequences of behavioral responses were more similar to each other than either was to wLH. In addition to changing the average probabilities of pairs of responses, wLH also altered longer sequences of outcomes within animals, reflected in decreased probability and length of successive escapes, as well as increased successive failures. Ketamine treatment partially recovered the probability and length of successive escapes, while a more prominent effect was observed on decreasing successive failures back to baseline levels (*Figure 2—figure supplement 1b*). Altogether, these analyses suggest that ketamine specifically restructures behavioral sequences that are altered by wLH learning.

The weaker LH paradigm, along with a large number of escapable foot shocks as the test stimuli, enabled broad sampling of Ca²⁺ transients during both escape and failure trials across conditions. Similar reductions in VTA DA responses to inescapable foot shocks were observed during wLH induction, as for strong LH (*Figure 2—figure supplement 1c*). For escapable shocks, separately plotting VTA DA neuron activity for trials where the animal escaped or failed to escape revealed that each behavioral response is associated with distinct Ca²⁺ transient shapes (*Figure 2c*). Failure trial transients were biphasic, where fluorescence decreased during the shock and increased afterwards. In contrast, monophasic transients accompanied escape trials, with an increase in fluorescence following successful transitions to the other side of the box. After learning, an increase in the similarity of the

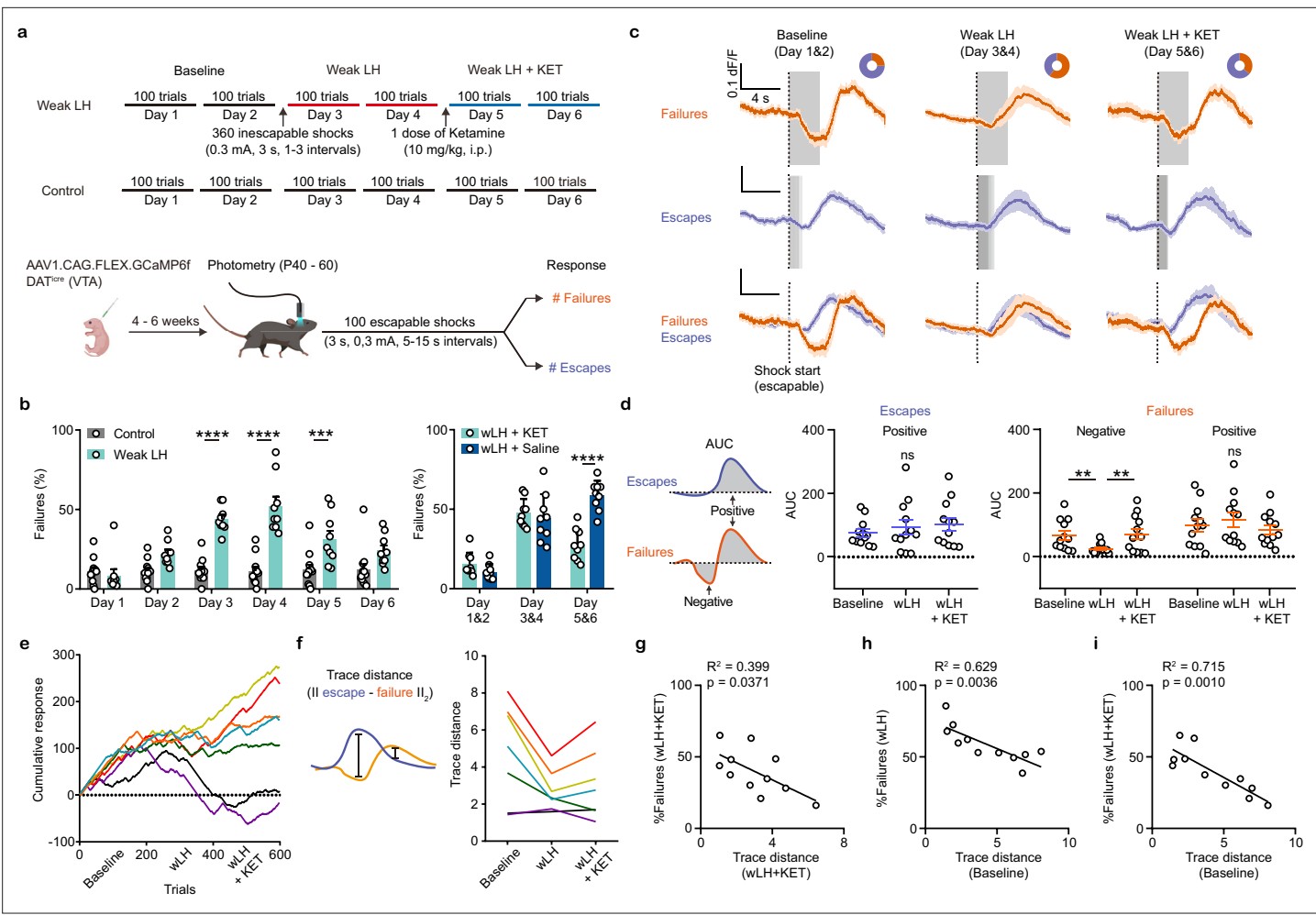

**Figure 2.** Weak learning analysis links DA activity, behavioral outcomes, and response to ketamine. (**a**). Top, schematic of experimental timeline for weak LH (wLH). Bottom, schematic of viral transduction, test trial description, and timing of photometry recording. (**b**). Left, summary data showing the percentage of failures to escape an escapable aversive shock across 6 days for two groups (Gray bar, controls; Cyan, wLH). Right, summary of behavioral data for wLH mice with KET or Saline treatment. Left, responses across days, two-way ANOVA, Sidak's multiple comparison test, Control vs wLH, Day 1, p = 0.9934, Day 2, p = 0.1147, Days 3 and 4, p < 0.0001, Day 5, p = 0.0007, and D6, p = 0.0691. Control, n = 12; wLH, n = 9 animals. Right, two-way ANOVA, Sidak's multiple comparison test, wLH+ KET vs wLH+ Saline, Days 1 and 2, p = 0.6403, Days 3 and 4, p = 0.8715, Days 5 and 6, p < 0.0001. n = 9 animals/group. (**c**). Fiber photometry recordings of VTA DA $Ca^{2+}$ transients, separated by behavioral response and aligned to shock start time (purple, escape; orange, failure; average dF/F across animals). Gray rectangles mark shock length in time, which is constant for failures to escape but variable for successful escapes and is shaded proportionally. Donut plots depict proportion of behavioral responses. n = 8 animals, baseline: 1405 trials, 25% failure 75% escape; wLH: 1459 trials, 60% failure 40% escape; wLH+ KET: 1403 trials, 36% failure, 64% escape. (**d**). Left, schematic illustration for measured variables. Middle, summary data for AUC of the positive $Ca^{2+}$ transient peaks in escapes across learning phases. Right, same but AUC for both positive and negative peaks in failures, as shown in the schematic. n = 12 animals, AUC (Escapes), positive peak, repeated measures one-way ANOVA, $F_{(1.897, 20.87)}$ = 1.881, p = 0.1787. AUC (Failures), negative peak, repeated measures one-way ANOVA, $F_{(1.852, 20.38)}$ = 9.260, p = 0.0017, Holm-Sidak's multiple comparison test, Baseline vs wLH, p = 0.0069, wLH vs wLH+ KET, p = 0.0069. Positive peak, repeated measures one-way ANOVA, $F_{(1.540, 16.94)}$ = 2.541, p = 0.1181. (**e**). Learning curves for individual animals. n = 7 animals, 600 trials/animal. (**f**). Left, schematic illustration for the measured variable, Euclidean norm of the difference between mean escape and failure-associated $Ca^{2+}$ transients for each subject. Right, trace distances for each subject across learning phases with subject specific colors as in e. (**g**). Correlation between trace distance and the percentage of failures in wLH+ KET. n = 11 animals. (**h**). Correlation between trace distance in the baseline condition and the percentage of failures in wLH. n = 11 animals. (**i**). Correlation between trace distance in the baseline condition and the percentage of failures in wLH+ KET. n = 11 animals. ** p < 0.01, *** p < 0.001, **** p < 0.0001. Error bars reflect SEM.

The online version of this article includes the following figure supplement(s) for figure 2:

**Source data 1.** Numerical data for the graphs in *Figure 2*.

**Figure supplement 1.** Characterization of behavioral sequences and dampened VTA DA activity in weak learned helplessness (wLH) paradigm.

**Figure supplement 1—source data 1.** Numerical data for the graphs in *Figure 2—figure supplement 1*.

*Figure 2 continued on next page*

*Figure 2 continued*

**Figure supplement 2.** Outcome-specific VTA DA responses in wLH paradigm.

**Figure supplement 2—source data 1.** Numerical data for the graphs in *Figure 2—figure supplement 2*.

**Figure supplement 3.** Outcome-specific VTA DA responses in control mice without wLH induction.

**Figure supplement 3—source data 1.** Numerical data for the graphs in *Figure 2—figure supplement 3*.

**Figure supplement 4.** Outcome-specific VTA DA responses in wLH mice with saline treatment.

**Figure supplement 4—source data 1.** Numerical data for the graphs in *Figure 2—figure supplement 4*.

**Figure supplement 5.** Trace distance of GCaMP transients between escape and failure trials.

activity patterns between escape and failure trials was observed (*Figure 2c and d*). For successful escape responses, VTA DA Ca$^{2+}$ transients did not change in wLH and wLH+ ketamine. The increased similarity of Ca$^{2+}$ transients between outcomes after learning was explained by the reduction in negative transients associated with failures. These failure-linked negative transients recovered following ketamine treatment (*Figure 2d*). The timing of peaks in failure outcomes varied across conditions (*Figure 2—figure supplement 2a*, b). Outcome-specific GCaMP6f transients, for either successful escapes or failures, did not vary across days in control mice without wLH induction, suggesting that the exposure to inescapable shocks is necessary to shape the responses of VTA DA neurons (*Figure 2— figure supplement 3a*, b). In a separate control group, saline treatment did not change DA transients in either escape or failure trials after wLH (*Figure 2—figure supplement 4a*, b).

In humans and other animals, individual variability in stress susceptibility and in responsiveness to neuroactive compounds is broadly acknowledged (*Honey et al., 2008*). To visualize behavioral trajectories, learning curves were constructed by starting at 0 and incrementing by one for every escape trial, decrementing by one for every failure, and keeping constant for spontaneous transitions (*Figure 2e*). These learning curves highlight individual variability in learning and ketamine responsiveness. To depict the relationship between an animal's ketamine responsiveness and VTA DA activity, we plotted the Euclidean norm of the difference between their Ca$^{2+}$ transients during escape and failure trials (‖*escape − failure*‖$_2$) across learning (*Figure 2f*, *Figure 2—figure supplement 5a*, b). We selected this trace distance metric because it is agnostic to amplitude and kinetics of the response, specifically assaying the degree to which the activity of VTA DA neurons differentiates specific trial outcomes. *Figure 2g* shows a correlation between behavioral response (% failures) and trace distance following ketamine. Stronger ketamine behavioral effects were linked to more distinct Ca$^{2+}$ transients when comparing escape and failure trials (R$^2$ = 0.399). Animals characterized by a larger trace distance in the baseline were more resilient to inescapable stress during LH illustrated by fewer failures to escape after wLH (*Figure 2h*, R$^2$ = 0.629). Intriguingly, individuals with larger differences between DA GCaMP signals on escape and failure trials in the baseline showed stronger responses to future ketamine treatment (*Figure 2i*, R$^2$ = 0.715).

## DA neuron activity is necessary for behavioral effects of ketamine after LH

The association between DA transients, behavioral states, and trial outcomes raises the possibility that particular signatures of DA signals in response to aversive stimuli are required for escape actions. To test whether VTA DA activity is required for escape actions in general, or alternatively, whether it becomes important for restoring escapes after LH, we conditionally expressed an inhibitory DREADD (designer receptor exclusively activated by designer drug) hM4Di in VTA DA neurons. This engineered muscarinic receptor is activated by clozapine-N-oxide (CNO) to drive Gα$_i$-coupled pathways (*Armbruster et al., 2007*; *Kozorovitskiy et al., 2015*; *Kozorovitskiy et al., 2012*; *Roth, 2016*). We used a CBA promoter-driven Cre-conditional hM4Di AAV, with expression in VTA DA neurons confirmed by immunofluorescence (*Figure 3a*). Efficacy was evaluated using cell-attached recordings of hM4Di-mCherry$^+$ VTA DA neurons, with a flow in and a washout of 1 µM CNO (*Figure 3b*).

Mice conditionally expressing hM4Di in VTA DA neurons were compared to their Cre$^-$ littermate controls in the strong LH paradigm. Here, two ketamine treatments were administered on sequential days—the first one along with CNO, and the second one without (*Figure 3c*). The expression of hM4Di in DA neurons did not change baseline behavior or learned helplessness induction (*Figure 3d*). However, inhibiting DA neurons by CNO application (3 mg/kg, b.w., i.p.) co-administered with

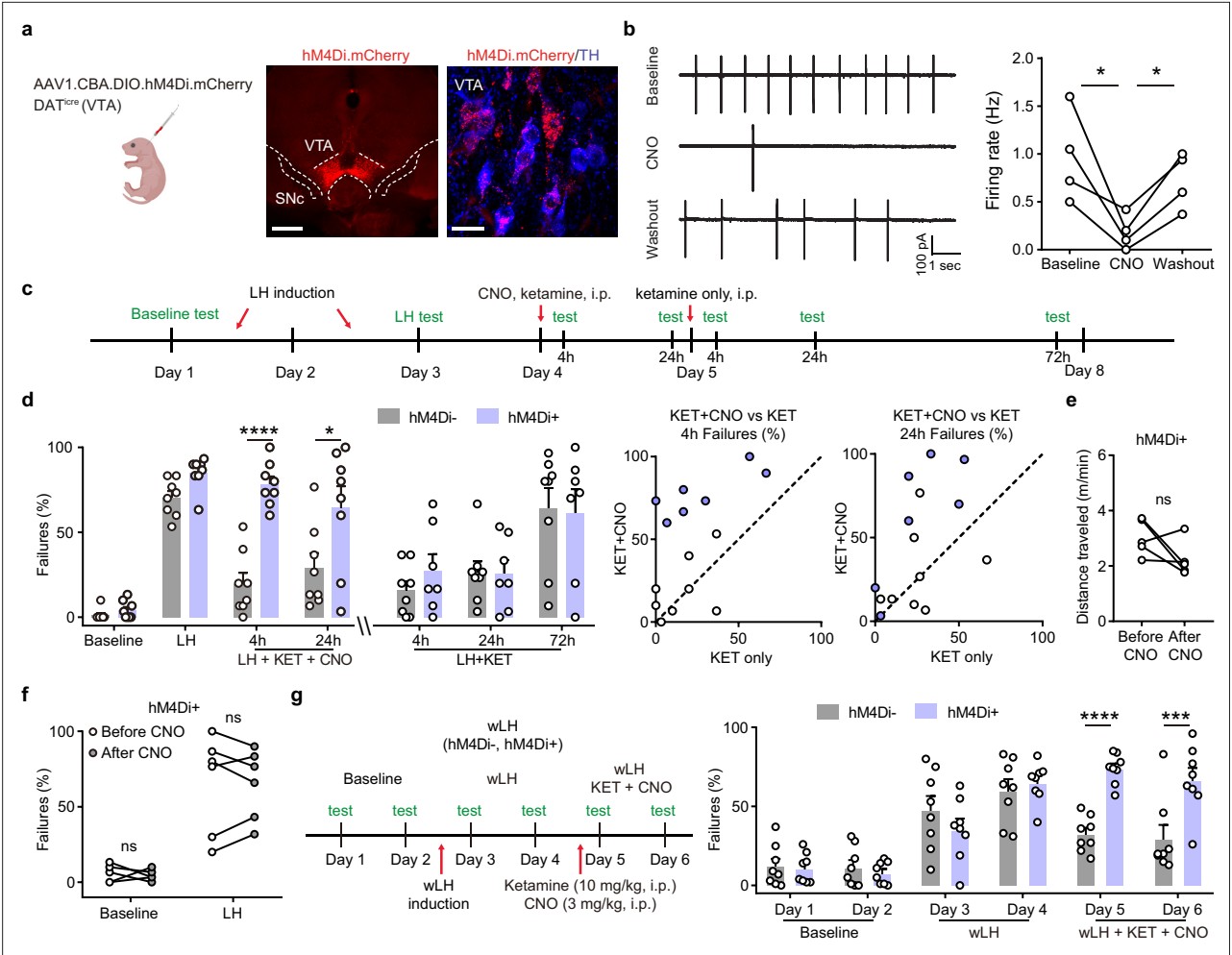

**Figure 3.** Ketamine behavioral effects in aversive learning require VTA DA activity. (**a**). Schematic for viral transduction in the VTA and hM4Di expression in VTA DA neurons (red, hM4Di.mCherry; blue, tyrosine hydroxylase (TH) immunofluorescence). Scale bars, 500 μm and 20 μm. (**b**). Cell-attached recording of spontaneous activity in mCherry[+] VTA neuron before, during, and after bath application of 1 μM clozapine-N-oxide (CNO). Example traces (left) and summary data (right). n = 4 cells from three mice, repeated measures one-way ANOVA, $F_{(2, 6)}$ = 9.634, p = 0.0134, Sidak's multiple comparison test, Baseline vs CNO, p = 0.0104, CNO vs Washout, p = 0.0488. (**c**). Schematic of experimental timeline and pharmacological interventions in strong LH paradigm. (**d**). Left, summary data showing the percentage of failures to escape an escapable aversive shock across learning and treatment conditions for hM4Di AAV expressing DAT[iCre]-positive and negative littermates. Middle, within subject summary data for behavioral responses after ketamine treatment only, compared with ketamine+ CNO, 4 hr after treatment. Right, same but for 24 hr after treatment. Two-way ANOVA, Sidak's multiple comparison test, KET + CNO 4 hrs, p < 0.0001, KET + CNO 24 hr, p = 0.0107, KET only, following 4, 24, and 72 hr, p > 0.9, n = 7–8 animals. (**e**). Total distance traveled per minute in an open field locomotion assay for hM4Di[+] animals before and after CNO treatment. Two-tailed paired t-test, p = 0.1473, n = 5 animals. (**f**). Summary data showing the percentage of failures to escape an escapable aversive shock in the baseline condition, for hM4Di-expressing mice before and after CNO treatment. n = 5–6 animals, Two-way ANOVA, Sidak's multiple comparison test, Baseline, p = 0.9538, LH, p = 0.9947. (**g**). Left, schematic of experimental timeline and pharmacological interventions in weak LH paradigm. Right, summary data showing the percentage of failures to escape an escapable aversive shock across 6 days for two groups (white bar, hM4Di- controls; purple, hM4Di+). Two-way ANOVA, Sidak's multiple comparison test, hM4Di + vs hM4Di-, D1-D4, p > 0.5, D5, p < 0.0001, D6, p = 0.0001, n = 8 animals. *p < 0.05, *** p < 0.001, **** p < 0.0001. Error bars reflect SEM.

The online version of this article includes the following figure supplement(s) for figure 3:

**Source data 1.** Numerical data for the graphs in *Figure 3*.

ketamine (10 mg/kg, b.w., i.p.) blocked ketamine's rescue of escape behaviors. This effect was evident 4 hr after CNO/Ketamine treatment and persisted at 24 hr. Yet, when ketamine was administered alone on the following day, the behavioral rescue in hM4Di[+] mice was successful. Persistent LH was evident in multiple animals of both groups 72 hr following the ketamine only treatment (last two bars, *Figure 3d*). Comparing behavioral responses following ketamine alone versus ketamine plus CNO,

within animals at two separate time-points, showed that the responses of hM4Di⁻ animals lie around the unity line (**Figure 3d**). In contrast, hM4Di⁺ responses were above the unity line, reflecting selective efficacy of ketamine treatment in the absence of CNO. Locomotor behavior in hM4Di-expressing animals was not grossly altered by a single CNO administration, suggesting that the observed changes in escape actions are not due to reduced movement (**Figure 3e**).

Although chemogenetic inhibition of DA activity blocked the behavioral effect of ketamine after aversive learning, the administration of CNO in the absence of ketamine in separate groups of hM4Di⁺ mice did not alter the proportion of failures to escape in the baseline condition or after sLH (**Figure 3f**). These data support the alternative hypothesis that the effect of ketamine after LH specifically requires the associated restoration of DA signals. Although VTA DA activity varies with behavioral outcomes before LH, innate escape actions may not fundamentally require DA activity. In addition, CNO co-administration blocked the behavioral effects of ketamine in the relatively weaker learning paradigm, as expected (wLH, **Figure 3g**). These data support the necessity of VTA DA signaling for the behavioral reversal of aversive learning by ketamine, rather than for native escape responses in naïve animals.

## Ex vivo local effects of ketamine on VTA DA neurons

In the next series of experiments, we sought to address the mechanisms underlying ketamine modulation of VTA DA activity and escape actions. In vivo ketamine treatment may modulate DA neuronal activity in a cell-autonomous manner locally in the VTA, or alternatively, indirectly, through effects on brain regions interconnected with the VTA. To determine whether ketamine application changes VTA DA GCaMP activity locally, we performed two-photon calcium imaging of VTA DA neurons with and without ex vivo ketamine application. DAT^icre neonates were transduced with AAV1.CAG.FLEX. GCaMP6f, and acute brain slices of VTA were prepared and imaged 4–6 weeks after viral transduction (**Figure 4a and b**). We observed spontaneous Ca²⁺ oscillations of DA neurons ex vivo (**Figure 4c**), which have been shown to match action potentials in previous reports (**Engelhard et al., 2019**). Further, bath application of ketamine (50 µM) in acute VTA brain slices did not alter the power or frequency tuning of spontaneous Ca²⁺ dynamics in DA neurons (**Figure 4C**), suggesting that the effect of in vivo ketamine treatment on VTA DA Ca²⁺ dynamics is not likely to be caused by direct cell-autonomous regulation. To reveal whether ex vivo ketamine application regulates VTA DA firing or spontaneous synaptic inputs, we also carried out cell-attached and whole-cell voltage clamp recordings from genetically targeted VTA DA neurons. DAT^icre neonates were transduced with AAV1.FLEX. EGFP, and EGFP-expressing cells were recorded in acute brain slices from the VTA (**Figure 4g**). Consistent with imaging results, ex vivo application of ketamine did not change spontaneous firing rate of VTA DA cells (**Figure 4h and i**), or the amplitudes and inter-event intervals of spontaneous EPSCs and IPSCs (**Figure 4J-o** ). Altogether, our data suggest the in vivo effects of ketamine on DA activity and behavior are not likely caused by local regulation in the VTA.

## Local mPFC ketamine infusion rescues VTA Ca²⁺ transients and escape actions after LH

The absence of local modulation of DA neuron activity by ketamine supports the possibility of circuit-level mechanisms. In addition to spontaneous tonic activity, VTA DA neurons are driven by multiple sources of glutamatergic excitation, including inputs from mPFC, PPTg, and PAG, among others (**Morales and Margolis, 2017**). To explore whether ketamine acts through one or more among core glutamatergic VTA inputs to rescue escape actions, we locally infused ketamine into three different brain regions (mPFC, PAG, and PPTg), along with the VTA itself, in separate experiments (**Figure 5a**). Local infusion of ketamine into mPFC alone sufficed to rescue escape behavior after LH (**Figure 5b**, **Figure 5—figure supplement 1a**). In contrast, ketamine infusion into the periaqueductal gray (PAG) and the pedunculopontine tegmental nucleus (PPTg) did not rescue escape actions. Local VTA infusion of ketamine also failed to rescue escape behavior after LH, further supporting the idea that the in vivo effects of ketamine on behavior are not implemented through local regulatory mechanisms in the VTA (**Figure 5b**).

To test whether local ketamine effects in mPFC rescue VTA DA dynamics after aversive learning in parallel to the behavioral changes, we infused ketamine locally into mPFC (12.5 µg, 500 nl), with fiber photometry as a readout of VTA activity (**Figure 5c**). In vivo local infusion of ketamine into mPFC sufficed to recover Ca²⁺ activity signatures in VTA DA neurons after aversive learning, along with

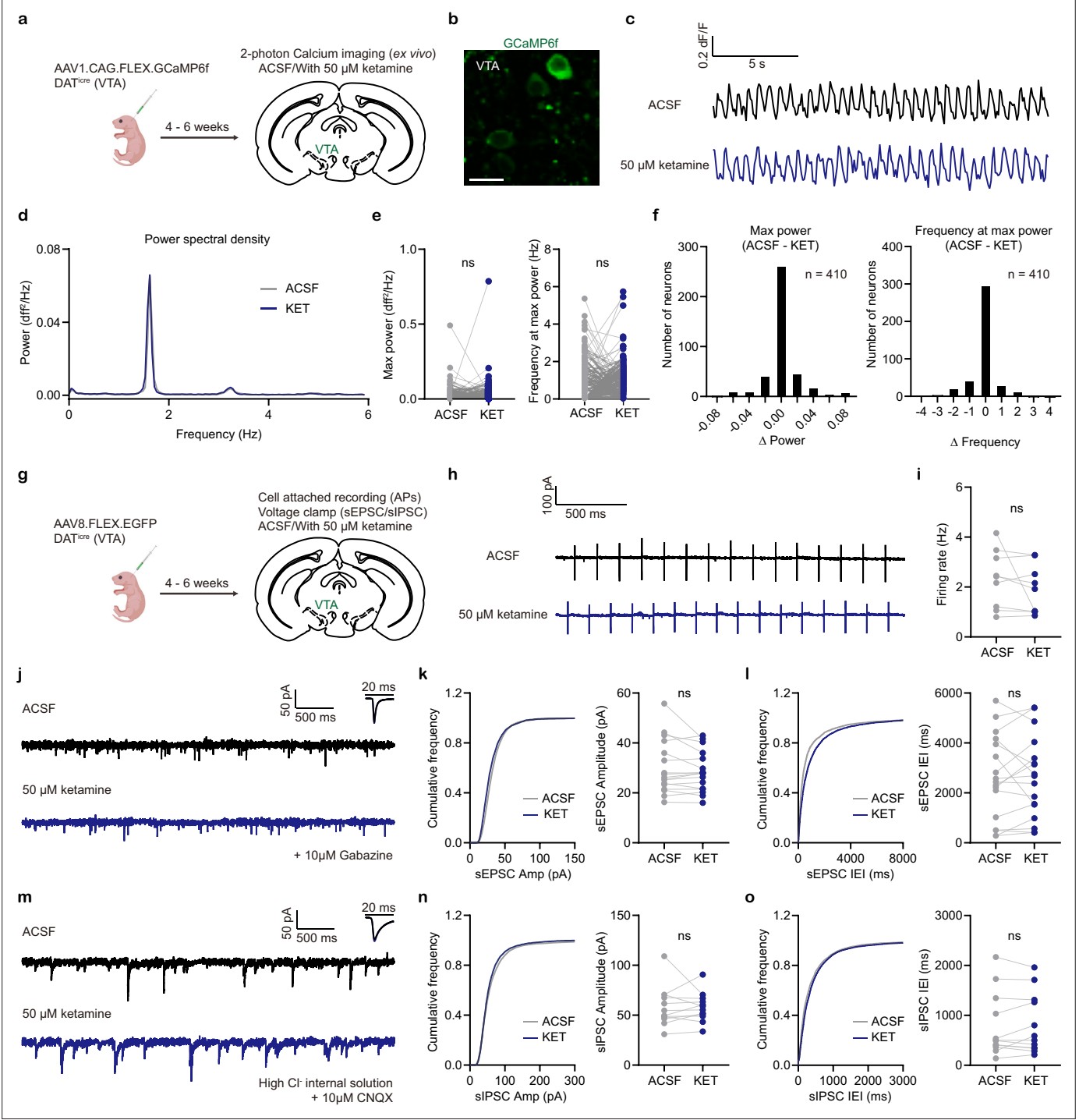

**Figure 4.** Ex vivo ketamine application does not alter GCaMP6f transients, neuronal firing, and synaptic inputs of VTA DA neurons. (**a**). Schematic illustrating viral transduction strategy and two-photon Ca²⁺ imaging of VTA DA neurons in acute brain slices. (**b**). Example 2PLSM image of VTA DA neurons expressing GCaMP6f. Scale bar, 20 μm. (**c**). Spontaneous Ca²⁺ oscillations in one neuron with and without ketamine bath application (50 μM). Black, ACSF; blue, with ketamine. (**d**). Power spectral density of Ca²⁺ transients for the neuron in (**c**). (**e**). Left, quantification of max power with and without ketamine treatment (n = 410 neurons). Right, quantification of frequency at max power. Paired two-tailed t test, ACSF vs KET, Max power, p = 0.8865, Frequency at max power, p = 0.3779. (**f**). Left, histogram showing the distribution of changes in max power with ketamine application. Right, same but for frequency at max power. n = 410 neurons. (**g**). Schematic illustrating viral transduction strategy and electrophysiological recording of VTA DA neurons in acute brain slices. (**h**). Spontaneous action potentials recorded in one neuron with and without ketamine bath application (50 μM). Black, ACSF; blue, with ketamine. (**i**). Quantification of neuronal firing rate with and without ketamine treatment (n = 9 neurons from three animals). Paired two-tailed t test, ACSF vs KET, p = 0.2561. (**j**). Spontaneous EPSCs recorded in one neuron with and without ketamine bath application (50 μM). Black, ACSF;

*Figure 4 continued on next page*

*Figure 4 continued*

blue, with ketamine. Holding membrane potential at –70 mV, 10 μM Gabazine in ASCF for both conditions. (**k**). Left, cumulative frequency distribution of sEPSCs amplitudes. Right, quantification of sEPSC amplitude in recorded neurons. Paired two-tailed t test, ACSF vs KET, p = 0.1958. n = 17 neurons from three animals. (**l**). Same as (**k**), but for sEPSCs inter-event intervals (IEI). Paired two-tailed t test, ACSF vs KET, p = 0.8413. (**m**). Spontaneous IPSCs recorded in one neuron with and without ketamine bath application (50 μM). Black, ACSF; blue, with ketamine. Holding membrane potential at –70 mV with high chloride internal solution, 10 μM CNQX in ASCF for both conditions. (**n**). Left, cumulative frequency distribution of sIPSCs amplitudes. Right, quantification of sIPSC amplitude in recorded neurons. Paired two-tailed t test, ACSF vs KET, p = 0.9164. n = 12 neurons from two animals. (**o**). Same as (**n**), but for sIPSCs inter-event intervals (IEI). Paired two-tailed t test, ACSF vs KET, p = 0.5675.

The online version of this article includes the following figure supplement(s) for figure 4:

**Source data 1.** Numerical data for the graphs in *Figure 4*.

behavioral rescue (*Figure 5d*), while infusion of ACSF did not recover VTA DA activity (*Figure 5—figure supplement 1b*, c). These data support the idea that ketamine rescues VTA DA dynamics through circuit-level effects involving mPFC.

## mPFC Drd1+ neurons mediate behavioral effects of ketamine

Next, in order to define the mPFC cell populations that project to the VTA, we combined retrograde tracing with fluorescence in situ hybridization (FISH). We injected red retrograde tracer fluorescent beads (retrobeads, RTB) into the VTA (*Figure 6a*) and observed red RTB fluorescence in mPFC 7–9 days after injection (*Figure 6b*). Results from FISH experiments demonstrate that the majority of RTB⁺ neurons in deep layers (layers 5/6) of mPFC express *Drd1a* mRNA. In contrast, only a small fraction of the retrobead⁺ neurons in layers 2/3 of mPFC have *Drd1a* mRNA (*Figure 6b and c*). To determine whether the deep layer Drd1⁺ population in mPFC projects to the VTA, we expressed AAV8.FLEX.EGFP in Drd1^Cre(FK150) animals and found that VTA neurons reside within fields of dense projections from mPFC Drd1⁺ neurons (*Figure 6d*). These anatomical tracing results are consistent with previous reports showing that VTA-targeting mPFC neurons may also receive VTA DA inputs (*Beier et al., 2015*; *Morales and Margolis, 2017*).

To determine whether ketamine modulates the activity of DA-sensing mPFC Drd1⁺ neurons in vivo, we used fiber photometry to monitor the activity of Drd1⁺ neurons in mPFC before and after ketamine treatment. To restrict the expression of GCaMP6f, we transduced AAV1.CAG.FLEX.GCaMP6f in Drd1^Cre(FK150) mice (*Figure 6—figure supplement 1a, b*). We observed a rapid enhancement in population activity of cortical Drd1⁺ neurons after i.p. ketamine treatment (*Figure 6—figure supplement 1c, d*). This ketamine-induced enhancement of population activity is likely mediated by suppressing the activity of local inhibitory interneurons (*Ali et al., 2020*; *Hare et al., 2020*; *Homayoun and Moghaddam, 2007*). If ketamine effect on mPFC Drd1⁺ neurons is crucial for behavioral changes, then the activation of mPFC Drd1⁺ neurons should rescue escape actions. We used the Gα_q-coupled hM3D to directly enhance the excitability of Drd1 expressing neurons in mPFC following aversive learning (*Figure 6e*). The expression and functional activation of hM3Dq were validated by immunohistochemistry and electrophysiology (*Figure 6f*). The expression of hM3Dq also did not change baseline escape/failure rates, or the magnitude of aversive learning. After LH induction, a single i.p. dose of CNO was sufficient to rescue escape behavior rapidly within 2 hr, with a larger effect observed 4 hr after CNO treatment (*Figure 6g*). At the end of the experiment, the successful enhancement of neuronal activity was further validated by evaluating immediate early gene product c-fos expression in mCherry⁺ neurons (*Figure 6h*). The activation of hM3Dq in Drd1-positive mPFC neurons did not alter locomotion, suggesting the rescue of escape action is not due to hyperlocomotor activity (*Figure 6i*). In addition to our results, a recently published study showed that optogenetic activation of Drd1⁺ mPFC neurons decreases immobility time in the forced swim test, suggesting the possibility that these Drd1-expressing neurons may broadly regulate aversive or active coping responses (*Hare et al., 2019*). Furthermore, consistent with prior findings (*Hare et al., 2019*), chemogenetic inhibition of mPFC Drd1⁺ neuronal excitability through Gα_i-coupled hM4D blocked the behavioral effects of ketamine after LH (*Figure 6—figure supplement 2a,b*). Altogether, our data demonstrate that mPFC serves as an action site of ketamine to rescue VTA DA dynamics, which is necessary to drive escape actions after LH.

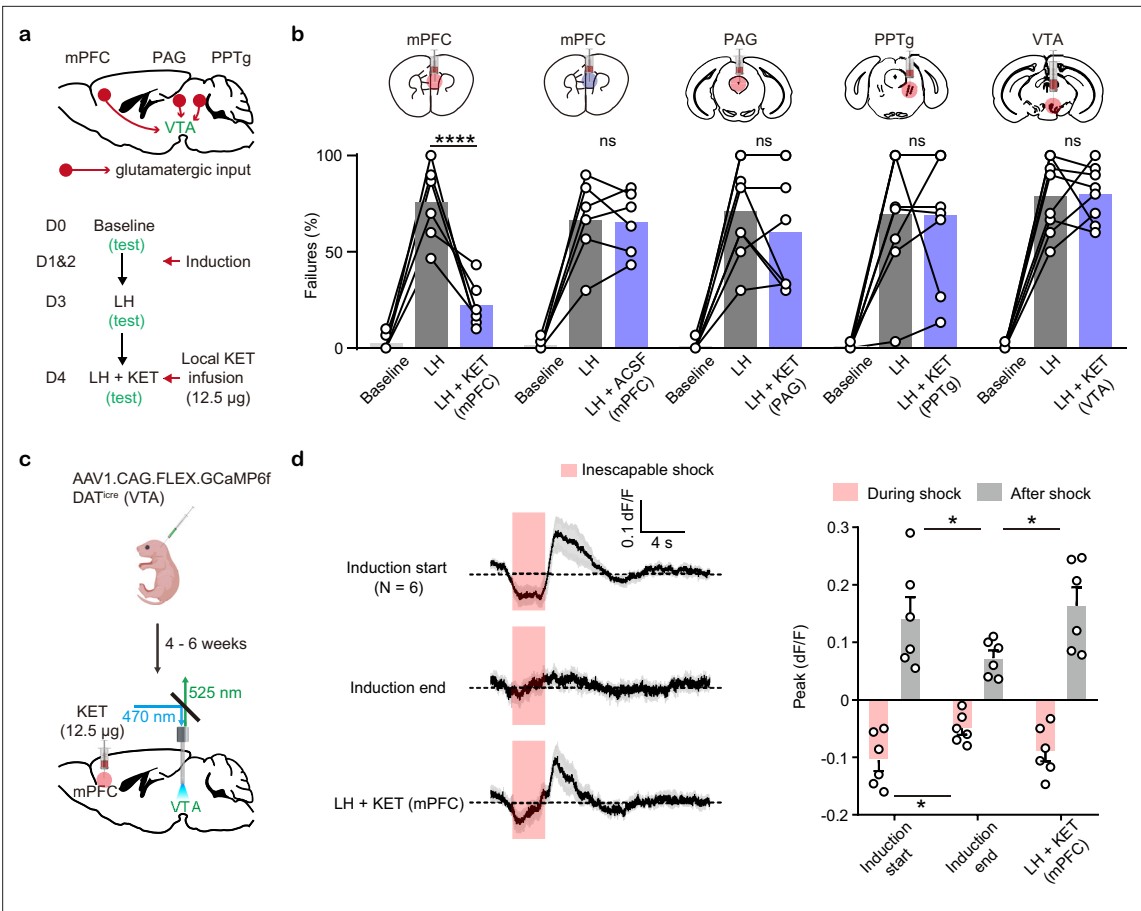

**Figure 5.** Local ketamine infusion in mPFC rescues escape behavior and VTA DA activity after LH. (**a**). Top, schematic for selected glutamatergic input regions to VTA. Bottom, experimental timeline. (**b**). Summary data showing the percentage of failures after local infusion of ketamine or ACSF in mPFC, and ketamine in PAG, PPTg, and VTA. Two-way ANOVA, Sidak's multiple comparison test, LH vs LH+ KET, mPFC, p< 0.0001, PAG, p = 0.4965, PPTg, p = 0.9998, VTA, p = 0.9986. LH vs LH+ ACSF (mPFC), p = 0.9993. (**c**). Schematic for viral transduction and VTA photometry recording of $Ca^{2+}$ transients with local ketamine delivery in mPFC. (**d**). Left, average $Ca^{2+}$ transients (mean ± SEM) in response to foot shocks at the start of induction, at the end of induction, and following local ketamine infusion. Traces are aligned to shock start time (20 trials/animal, 6 animals). Right, quantification of peak $Ca^{2+}$ transient amplitude during and after foot shock stimuli across conditions. Both positive and negative values are quantified. n = 6 animals, repeated measures one-way ANOVA, Holm-Sidak's multiple comparison test, Peak: During shock, F (1.948, 9.739) = 5.547, p = 0.0252, Induction start vs Induction end p = 0.0433, Induction end vs LH + KET, p = 0.0823. After shock, F (1.468, 7.341) = 10.05, p = 0.0105, Induction start vs Induction end, p = 0.0462, Induction end vs LH + KET, p = 0.0147. *p < 0.05, *** p < 0.001, **** p < 0.0001. Error bars reflect SEM.

The online version of this article includes the following figure supplement(s) for figure 5:

**Source data 1.** Numerical data for the graphs in *Figure 5*.

**Figure supplement 1.** VTA DA activity after local ASCF infusion in mPFC.

**Figure supplement 1—source data 1.** Numerical data for the graphs in *Figure 5—figure supplement 1*.

## Discussion

Animals must learn to avoid dangerous stimuli in order to survive, underscoring the ethological significance of aversive learning. However, overlearned responses can be maladaptive. Prolonged exposure to stressful events can lead to learned helplessness (*Chourbaji et al., 2005*; *Maier, 1984*; *Maier and Seligman, 2016*), where an individual fails to avoid unpleasant stimuli. Here, we demonstrate that aversive learning restructures neuronal activity in the VTA DA system, associated with a shift in behavioral outcome distribution. A single low-dose ketamine treatment suffices to temporarily normalize both DA signaling and behavior, with the efficacy of treatment correlating with VTA DA activity features prior to learning. Inhibition of somatic DA activity or consequences of DA release

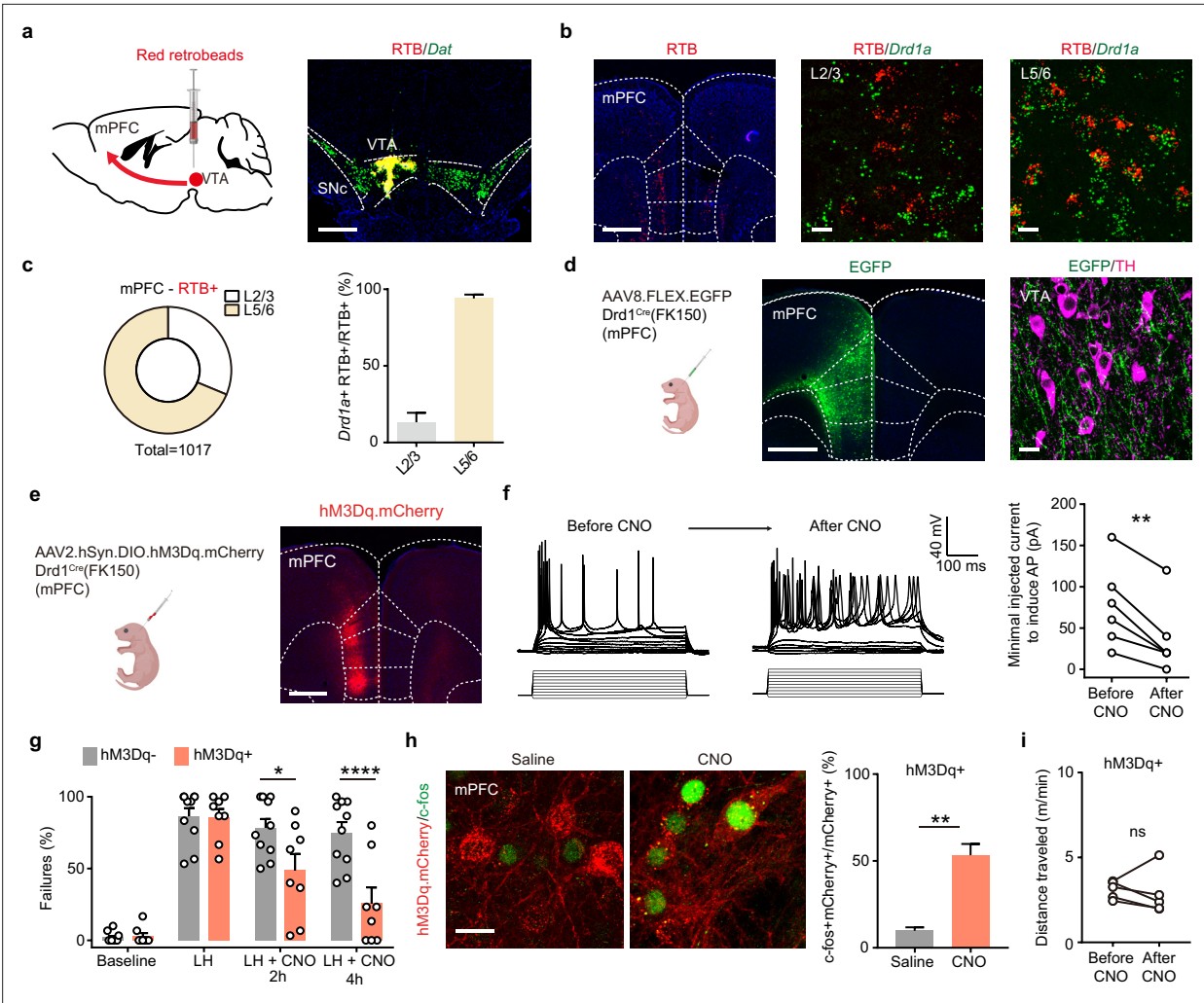

**Figure 6.** Activation of mPFC Drd1+ neurons rescues escape actions. (**a**). Left, schematic illustrating retrograde labeling strategy. Right, coronal image of retrobead (RTB) injection site. Red, retrobeads; green, *Dat* mRNA; blue, DAPI. Scale bar, 500 μm. (**b**). Left, retrobead fluorescence in mPFC (atlas overlay, dashed line). Middle and right, colocalization of RTB and *Drd1* mRNA in superficial and deep layers of mPFC. Scale bars, 500 μm (left), 20 μm (middle and right). (**c**). Left, the proportional distribution of RTB+ neurons across superficial and deep cortical layers. Right, quantification of the percentage of *Drd1a* + cells among RTB+ cells across layers. n = 2 animals, 31.3% RTB+ neurons in layer 2/3, 68.7% RTB+ in layer 5/6. Among those RTB+ neurons, 13.2% ± 6.2% are *Drd1*+ in layer 2/3, + ± 2.5 are *Drd1*+ in layer 5/6. (**d**). Left, schematic illustrating viral transduction strategy. Right, a coronal image showing the expression of EGFP (green) in the mPFC Drd1+ neurons. Scale bar, 500 μm. (**e**). Left, schematic illustrating viral transduction strategy. Right, epifluorescent image of hM3Dq-mCherry expression in mPFC. Scale bar, 500 μm. (**f**). Whole-cell recording of action potentials evoked by different amplitude current injections in one mCherry+ mPFC neuron before, during, and after bath application of 1 μM CNO. Left, example traces. Right, summary data for minimal injected current sufficient to evoke action potential firing before and after CNO application. n = 6 cells from two animals, two-tailed paired t-test, p = 0.0028. (**g**). Summary data showing the percentage of failures to escape an escapable aversive shock in Drd1-Cre+ and Drd1-Cre- mice expressing hM3Dq across phases of learning and after CNO treatment (Baseline, LH, LH+ CNO 2 hrs, and LH+ CNO 4 hrs). n = 8 animals for Cre-, n = 10 mice for Cre+, two-way ANOVA, Sidak's multiple comparison test, LH+ CNO 2 hr, p = 0.0157, LH+ CNO 4 hr, p < 0.0001, Baseline/LH, p > 0.9. (**h**). Left, colocalization of c-fos immunolabeling in hM3Dq.mCherry+ mPFC neurons in saline and CNO-treated Drd1 Cre+, hM3Dq-expresing mice. Right, the quantification of percentage of c-fos+ cells among mCherry+ cells. Scale bar, 20 μm. n = 3 mice/condition, two-tailed unpaired t-test, p = 0.0031. (**i**). Total distance traveled per minute in an open-field locomotion assay for hM3Dq+ animals before and after CNO treatment. Two-tailed paired t-test, p = 0.6289, n = 5 animals. *p < 0.05, ** p < 0.01, **** p < 0.0001. Error bars reflect SEM.

The online version of this article includes the following figure supplement(s) for figure 6:

**Source data 1.** Numerical data for the graphs in *Figure 6*.

**Figure supplement 1.** Activity of Drd1+ neurons in mPFC after in vivo ketamine treatment.

**Figure supplement 1—source data 1.** Numerical data for the graphs in *Figure 6—figure supplement 1*.

**Figure supplement 2.** Inhibition of Drd1+ neurons in mPFC blocks the behavioral effects of ketamine.

*Figure 6 continued on next page*

*Figure 6 continued*

**Figure supplement 2—source data 1.** Numerical data for the graphs in *Figure 6—figure supplement 2*.

**Figure supplement 3.** Behavioral outcomes and viral expression distributions.

**Figure supplement 3—source data 1.** Numerical data for the graphs in *Figure 6—figure supplement 3*.

abolishes the behavioral effects of ketamine. We identify mPFC as a key action site that mediates ketamine's effects on VTA DA dynamics, where the activation of mPFC Drd1$^+$ neurons is sufficient to rescue escape actions.

## Individual variability in VTA DA activity and behavioral susceptibility

Multivalenced encoding of information by VTA DA neurons and their projections is now well established (*Berridge and Kringelbach, 2015*; *Lammel et al., 2014*; *Morales and Margolis, 2017*; *Nestler and Carlezon, 2006*; *Schultz, 2016*). Prior reports demonstrate that stress and aversive learning change DA neuron activity in both acute and chronic paradigms (*Chaudhury et al., 2013*; *de Jong et al., 2019*; *Lammel et al., 2014*; *Lammel et al., 2012*; *Lammel et al., 2011*). Our findings here show that VTA GCaMP transients are shaped both by trial outcomes—escape or failure to escape from an avoidable stimulus—as well as by the general behavioral state induced by LH learning and ketamine treatment. This malleability and multifaceted activity of DA signaling opens the possibility that particular patterns and levels of VTA DA activity map to specific behavioral outcome distributions. In this framework, differences in VTA DA activity between success and failure outcomes map to the tendency to escape aversive stimuli. Accordingly, decreases in the differences between VTA DA activity on escape and failure trials after aversive learning are associated with increased failures to escape avoidable stressful stimuli. Future behavioral responses to ketamine are correlated with trace distances between escape and failure-associated GCaMP transients before learning occurs. Building on these correlative observations, chemogenetic inhibition of VTA DA neuron activity supports a causal relationship between learning, DA activity, and ketamine effects. These data demonstrate that DA activity changes in LH paradigms and perturbations of DA dynamics shape behavioral outcomes in escape behavior.

The question of what key biological parameters distinguish resilient and susceptible animals remains enigmatic. In this study, we observe substantial individual variability of behavioral expression after LH. Other studies have mapped this type of variability to a binary structure of resilience and susceptibility (*Bagot et al., 2017*; *Chaudhury et al., 2013*; *Friedman et al., 2014*; *Peña et al., 2019*; *Wang et al., 2017*). While there is some utility in forcing a binary behavioral outcome, in this relatively large dataset (n > 130 mice), we noted a broad (and not bimodal) variance in LH outcomes for both strong and weaker LH, reflecting the diversity of individual learning experiences (*Figure 6—figure supplement 3a*, b). Notably, we also observe biologically significant variability in VTA DA responses, correlated with future behavioral outcomes after learning and ketamine treatment. The biological nature of this variability may relate to resilience or susceptibility to stress. One interesting possibility is that projections and activity profiles of DA neuron subgroups could be related to early-life and adolescent experience (*O'Donnell, 2010*; *Peña et al., 2019*; *Peña et al., 2017*; *Wahlstrom et al., 2010*), resulting in variability of the overall low-dimensional activity pattern of the VTA DA population recorded using photometry. Based on this observation, it is expected that task-specific VTA DA activity may serve as a potential biomarker for the degree of susceptibility to stressful events. In the context of human populations, PET studies have been used to measure aspects of DA system function (*Jönsson et al., 1999*; *MacDonald et al., 2012*; *Okubo et al., 1997*; *Volkow et al., 1996*). Importantly, genetic variability which can drive differences in DA signaling (*Jönsson et al., 1999*), could provide additional biomarkers to predict clinical ketamine efficacy.

Animals used in this study derive from a genetically homogenous background, and one can expect larger variances in DA-associated genes in the human population. DA signaling has been associated with positive and negative emotionality and related to variability in psychopathology (*Felten et al., 2011*). Polymorphisms in DA genes regulate DA transmission throughout the brain, influencing depression-related phenotypes (*Frisch et al., 1999*; *Haeffel et al., 2008*; *Pearson-Fuhrhop et al., 2014*). Variance in DA genes and their interactors, along with early life experience, may shape activity

patterns of DA neurons and DA release in response to stressful events, defining depression susceptibility in humans.

## Circuit-level mechanisms underlying the effects of ketamine on behavior

Multiple studies demonstrate that mPFC is broadly involved in the rapid and lasting behavioral effects of ketamine. NMDA receptor antagonists, including ketamine, have been shown to rapidly enhance neuronal and dendritic activity of cortical pyramidal neurons through disinhibition (*Ali et al., 2020*; *Hare et al., 2020*; *Homayoun and Moghaddam, 2007*; *Picard et al., 2019*; *Zanos and Gould, 2018*). Further, on a relative longer timescale, increases in cortical dendritic spine density, induced by ketamine, maintain behavioral effects after ketamine clearance in vivo (*Li et al., 2010*; *Moda-Sava et al., 2019*). The enhancement in cortical spinogenesis requires mTORC1, BDNF, subtypes of glutamate receptors, as well as dopaminergic signaling (*Ali et al., 2020*; *Duman et al., 2012*; *Li et al., 2010*; *Liu et al., 2013*; *Sarkar and Kabbaj, 2016*; *Wu et al., 2021a*). Optogenetic stimulation of mPFC neurons reverses depressive-like behaviors (*Covington et al., 2010*; *Fuchikami et al., 2015*; *Kumar et al., 2013*). Evidence from this study and others has consistently demonstrated that the activation of mPFC Drd1 neurons is important for ketamine's behavioral effects. Consistent with diverse outputs of mPFC, local ketamine actions there may subsequently change the activity of multiple downstream brain regions, including the VTA, but also basolateral amygdala, nucleus accumbens (NAc), and lateral habenula, among others. These brain regions likely contribute to specific behavioral effects of ketamine observed in different behavioral paradigms (*Belujon and Grace, 2014*; *Hare et al., 2019*; *Kumar et al., 2013*; *Yang et al., 2018a*).

Besides modulating mPFC Drd1+ neurons, DA signaling in many brain regions regulates escape actions and avoidance behaviors. For example, the release and postsynaptic function of DA in the NAc and amygdala have been extensively studied in conditioned avoidance learning (*Antunes et al., 2020*; *Darvas et al., 2011*; *McCullough et al., 1993*; *Oleson et al., 2012*; *Oleson and Cheer, 2013*; *Stelly et al., 2019*; *Wenzel et al., 2018*). In this study, we found that VTA DA activity is not necessarily critical for innate escape actions, since chemogenetic suppression of VTA DA activity does not increase failures to escape from shock stimuli in the absence of prior learning. Importantly, after LH, the responses of VTA DA neurons to both inescapable and escapable shocks are significantly changed. Since DA release has been shown to be highly correlated with somatic DA neuronal activity (*de Jong et al., 2019*; *Lee et al., 2021*; *Patriarchi et al., 2018*), we predict that downstream DA release in the NAc and amygdala is also modulated after LH and ketamine treatment.

Which specific projections carry information about different aversive stimuli (e.g., conditioned cues or unpredicted aversive events) and whether they derive from the same or different sub-populations of VTA DA neurons, remains an active area of research (*Beier et al., 2015*; *de Jong et al., 2019*; *Lammel et al., 2011*; *Lammel et al., 2012*; *Yang et al., 2018b*). VTA DA neurons are highly heterogenous based on their input/output anatomy and transcriptional profiles (*Beier et al., 2015*; *Morales and Margolis, 2017*). Prior studies have demonstrated that VTA DA neurons respond to foot shocks differently, based on the brain regions they project to (*de Jong et al., 2019*). In this study, we do not distinguish responses of DA neurons based on their projections or transcriptional profiles. Thus, the observed effects are sufficiently powerful to be seen on the background of a mixed DA population that maintains projection target diversity.

## Ketamine effects on DA activity and mPFC-VTA recurrent circuit hypothesis

Given the absence of local ketamine effects on VTA DA neurons, as shown in ex vivo calcium imaging and electrophysiological recordings, along with in vivo infusion experiments, the modulation of VTA DA activity by ketamine in vivo must depend on changes in the activity of projections from other brain regions. Based on evidence from electron microscopy reconstructions and anatomical tracing, VTA DA neurons that project to mPFC receive inputs from mPFC pyramidal neurons, forming excitatory connections (*Beier et al., 2015*; *Carr and Sesack, 2000*). Trans-synaptic tracing analyses and optogenetic assays provide evidence that mPFC pyramidal neurons send excitatory glutamatergic projections to VTA (*Beier et al., 2015*; *Lodge, 2011*; *Xiao et al., 2018*; *Yang et al., 2018b*). Prior data from our lab (*Xiao et al., 2018*) and others (*Beier et al., 2015*; *Carr and Sesack, 2000*; *Geisler et al.,*

*2007*) confirm that VTA DA neurons receive excitatory mPFC inputs. Glutamate release from mPFC terminals elicits excitatory responses in VTA DA neurons (*Gariano and Groves, 1988*; *Kumar et al., 2013*; *Tong et al., 1996*; *You et al., 2007*), driving synchronized activity across VTA and mPFC, as well as other limbic structures (*Kumar et al., 2013*). Although the mPFC and VTA are the not only brain regions that control escape behaviors, the enhanced activity within these two brain regions is likely one important mechanism underlying behavioral effects of ketamine. Our data show that local mPFC ketamine infusion rescues both VTA DA activity and behavior (*Figure 5*). In vivo ketamine treatment rapidly enhances Drd1 neuronal activity in mPFC (*Figure 6—figure supplement 1*) and chemogenetic activation of mPFC Drd1+ neurons rescues escape behaviors (*Figure 6g*). Moreover, in recently published work, we show that DA signaling through Drd1 activation is necessary for ketamine's action on restoring glutamate-evoked dendritic spinogenesis in mPFC (*Wu et al., 2021a*), which has been shown to maintain the lasting behavioral effects of ketamine after corticosterone treatment (*Moda-Sava et al., 2019*). Chemogenetic activation of $G\alpha_s$-coupled signaling cascades downstream of Drd1 activation in mPFC also rescues escape behavior after LH (*Wu et al., 2021a*). Based on the sum of these data across multiple laboratories and experimental modalities, we hypothesize that the initial enhancement of activity induced by ketamine in mPFC may be amplified within recurrent mPFC-VTA circuit, which may help extend ketamine behavioral effects beyond its in vivo bioavailability. Future studies are necessary to explicitly test this model.

Other potential circuit mechanisms for ketamine's enhancement of VTA DA activity may involve ventral hippocampal inputs to nucleus accumbens and lateral habenula projections to the VTA, both of which are modulated by stressful experience and ketamine (*Bagot et al., 2015*; *Belujon and Grace, 2014*; *Li et al., 2011*; *Pignatelli et al., 2020*; *Yang et al., 2018a*). One such potential mechanism could involve actions in the lateral habenula, which sends glutamatergic inputs to GABAergic neurons that suppress VTA DA activity (*Gonçalves et al., 2012*). Excitatory synapses onto VTA-projecting lateral habenula neurons are potentiated in rats in behavioral and genetic models of LH (*Li et al., 2011*). Ketamine could disinhibit VTA DA neurons by suppressing the activity of lateral habenula glutamatergic neurons (*Yang et al., 2018a*). Rapid and lasting behavioral effects of ketamine likely result from synergistic actions across many brain regions, involving multiple molecular pathways.

## Clinical implications in co-morbidity

Drugs that modulate the dopaminergic system represent first line therapies for a large number of diverse neurological and mental health conditions that are comorbid with major depressive disorder, including Parkinson's disease, schizophrenia, OCD, ADHD, and eating disorders (*Biederman et al., 1998*; *Corcos et al., 2000*; *Masellis et al., 2003*; *Mayeux et al., 1981*; *Siris, 2000*). The recent FDA approval of esketamine for the treatment of major depressive disorder is a major expansion of its clinical use. Our results on dopaminergic mediation of ketamine's effects on behavior and plasticity suggest the possibility of ketamine's differential clinical effects in patients receiving exogenous DA precursors (e.g. L-DOPA), re-uptake inhibitors, and dopaminergic receptor agonists and antagonists.

## Materials and methods

Key resources table Key resources table

| Reagent type (species) or resource | Designation | Source or reference | Identifiers | Additional information |
|---|---|---|---|---|
| Strain, strain background (*M. musculus*) | C57BL/6 | Jackson Laboratory | Cat#000664;RRID: IMSR_JAX:000664 | |
| Strain, strain background (*M. musculus*) | B6.SJL-*Slc6a3tm1.1(cre)Bkmn*/J (DAT^iCre) | Jackson Laboratory | Cat#006660;RRID: IMSR_JAX:006660 | |
| Strain, strain background (*M. musculus*) | B6.FVB(Cg)-*Tg(Drd1a-cre)FK150Gsat*/Mmucd (Drd1^Cre (FK150)) | Jackson Laboratory | RRID:MMRRC_036916-UCD | |
| Recombinant DNA reagent | AAV1.CAG.Flex.GCaMP6f.WPRE.SV40 | *Chen et al., 2013*, Dr. Douglas Kim | Addgene viral prep; # 100835-AAV1 | |

*Continued on next page*

*Continued*

| Reagent type (species) or resource | Designation | Source or reference | Identifiers | Additional information |
|---|---|---|---|---|
| Recombinant DNA reagent | AAV8.CAG.Flex.EGFP | UNC Vector Core, (Dr. Boyden) | N/A | |
| Recombinant DNA reagent | AAV1.CBA.DIO.hM4Di. mCherry | *Hou et al., 2016*, Packaged by Vigene, (Plasmid, Dr. Sabatini) | Addgene plasmid # 81,008 | |
| Recombinant DNA reagent | AAV2.hSyn.DIO.hM3Dq. mCherry | *Krashes et al., 2011*, Dr. Bryan Roth | Addgene viral prep; # 44361-AAV2 | |
| Antibody | Rabbit polyclonal anti-Tyrosine Hydroxylase | Millipore | Cat#AB152; RRID: AB_390204 | (1:1000) |
| Antibody | Mouse monoclonal anti-Tyrosine Hydroxylase | Abcam | Cat#AB129991; RRID: AB_11156128 | (1:1000) |
| Antibody | Rabbit polyclonal anti-RFP | Rockland | Cat#600-401-379; RRID:AB_2209751 | (1:500) |
| Antibody | Rabbit polyclonal anti-c-Fos | Synaptic systems | Cat# 226 003; RRID:AB_2231974 | (1:5000) |
| Chemical compound, drug | Ketamine | Vedco | 217-484-6; CAS: 1867-66-9 | |
| Chemical compound, drug | Clozapine-N-oxide | Sigma-Aldrich | C0823; CAS: 34233-69-7 | |
| Chemical compound, drug | Red retrobeads (RRBs) | Lumafluor | Cat#R180 | |
| Chemical compound, drug | SR95531 hydrobromide | Tocris | Cat#1262; CAS 104104-50-9 | |
| Chemical compound, drug | CNQX disodium salt | Tocris | Cat#1045; CAS 479347-85-8 | |
| Chemical compound, drug | Chicago Sky Blue 6B | Tocris | Cat#0846; CAS 2610-05-1 | |
| Commercial assay or kit | RNAscope Fluorescence Multiplex Assay | ACDBio | Cat No. 320,850 | |
| Commercial assay or kit | RNAscope Probe-Mm-*drd1a(Drd1)* | ACDBio | Cat No. 406491-C2 | |
| Commercial assay or kit | RNAscope Probe-Mm-*slc6a3(Dat)* | ACDBio | Cat No. 315,441 | |
| Software, algorithm | GraphPad Prism 7 | GraphPad | RRID: SCR_002798 | |
| Software, algorithm | FIJI | *Schindelin et al., 2012* | http://fiji.sc/; RRID: SCR_002285 | |
| Software, algorithm | MATLAB | MathWorks | RRID: SCR_001622 | |
| Software, algorithm | Toxtrac | *Rodriguez et al., 2018* | N/A | |
| Software, algorithm | Python | Python Software Foundation | RRID:SCR_008394 | |
| Software, algorithm | Clampfit 11.2 | Molecular Devices | SCR_011323 | |
| Other | Active/Passive Avoidance Shuttle Box | MazeEngineers | https://mazeengineers.com/portfolio/active-passive-avoidance-shuttle-box/#description | |
| Other | Raspberry Pi | Raspberry Pi Foundation | https://www.raspberrypi.org/ | |

## Mouse strains and genotyping

Animals were handled according to protocols approved by the Northwestern University Animal Care and Use Committee. Weanling and young adult male and female mice (postnatal days 40–80) were used in this study. Approximately equal numbers of males and females were used for every experiment. All mice were group-housed, with standard feeding, 12 h light and 12 hr dark cycle (6:00 or 7:00 lights on), and enrichment procedures. Littermates were randomly assigned to conditions. C57BL/6 mice used for breeding and backcrossing were acquired from Charles River (Wilmington, MA), and all other mouse lines were acquired from the Jackson Laboratory (Bell Harbor, ME) and bred in house.

B6.SJL-*Slc6a3*[tm1.1(cre)Bkmn]/J mice, which express Cre recombinase under control of the dopamine transporter promoter, are referred to as DAT[iCre] (*Bäckman et al., 2006*); B6.FVB(Cg)-*Tg(Drd1-cre) FK150Gsat*/Mmucd mice, which express Cre recombinase under control of the dopamine Drd1a

receptor promoter, are referred to as Drd1Cre(FK150) (*Gong et al., 2007*). All transgenic animals were backcrossed to C57BL/6 for several generations. Heterozygous Cre+ mice were used in experiments. Standard genotyping primers are available on the Jackson Lab website.

## Stereotactic injections and optic fiber implants

Conditional expression of target genes in Cre-containing neurons was achieved using recombinant adeno-associated viruses (AAVs) using the FLEX cassette or encoding a double-floxed inverted open-reading frame (DIO) of target genes, as described previously (*Kozorovitskiy et al., 2015*). For fiber photometry and ex vivo two-photon calcium imaging experiments in the VTA, DATiCre mice were transduced with AAV1.CAG.FLEX.GCaMP6f.WPRE-SV40 (1.33 × 10^13 GC/ml) from the UPenn viral core (Philadelphia, PA, a gift from the Genetically Encoded Neural Indicator and Effector Project (GENIE) and Douglas Kim; Addgene viral prep #100835-AAV1) (*Chen et al., 2013*) or AAV8.CAG.FLEX.EGFP (3.1 × 10^12 GC/ml, UNC vector core, Dr. Ed Boyden). For fiber photometry experiments in the mPFC, Drd1Cre(FK150) mice were transduced with AAV1.CAG.FLEX.GCaMP6f.WPRE.SV40 (1.33 × 10^13 GC/ml). For chemogenetic experiments in mPFC, Drd1Cre(FK150) mice were transduced with AAV2.hSyn.DIO.hM3Dq.mCherry (8.6 × 10^12 GC/ml, Addgene viral prep #44361-AAV2, Dr. Bryan Roth) (*Krashes et al., 2011*). For chemogenetic experiments in VTA, DATiCre mice were transduced with a custom built AAV1.CBA.DIO.hM4Di.mCherry (1.28 × 10^13 GC/ml, Vigene Biosciences, Rockville, MD, plasmid a gift from Dr. Bernardo Sabatini) (*Hou et al., 2016*). For retrograde labeling, mice were intracranially injected with Red Retrobeads (Lumafluor Inc) in VTA.

Neonatal viral transduction was carried out to minimize invasiveness and increase surgical efficiency (*Bariselli et al., 2016*; *He et al., 2018*; *Kozorovitskiy et al., 2015*; *Kozorovitskiy et al., 2012*; *Peixoto et al., 2016*). P3-6 mice were cryoanesthetized, received ketoprofen for analgesia and were placed on a cooling pad. Virus was delivered at a rate of 100 nl/min for up to 150–200 nl using an UltraMicroPump (World Precision Instruments, Sarasota, FL). Medial prefrontal cortex (mPFC) was targeted in the neonates by directing the needle immediately posterior to the eyes, 0.3 mm from midline, and 1.8 mm ventral to skin surface. Ventral tegmental area (VTA) was targeted in the neonates by directing the needle approximately ±0.2 mm lateral from Lambda and 3.8 mm ventral to skin surface. Coordinates were slightly adjusted based on pup age and size. Following the procedure, pups were warmed on a heating pad and returned to home cages.

For intracranial injections, mice were anesthetized with isoflurane (3 % for induction value, 1.5%–2% for maintenance) or ketamine:xylazine (100:12.5 mg/kg b.w.), received ketoprofen for analgesia, and were placed on a small animal stereotax frame (David Kopf Instruments, Tujunga, CA). AAVs or retrobeads were delivered through a pulled glass pipette at a rate of 100–150 nl/min using an UltraMicroPump (World Precision Instruments, Sarasota, FL). Ketamine, ACSF, or Chicago Sky Blue 6B were delivered intracranially at a rate of 150 nl/min using the UltraMicroPump. Injection coordinates for VTA, 2.8 mm posterior to bregma, 0.4 mm lateral, and 4.3–4.5 mm below the pia; for mPFC, 2.3 mm anterior to bregma, 0.4 mm lateral, and 1.3–1.6 mm below the pia; for PAG, 4.2 mm posterior to bregma, 0.1 mm lateral, and 2.2 below the pia; for PPTg, 4.5 mm posterior to bregma, 1.2 mm lateral, and 2.5 below the pia. Pipettes were held at the injection location for 15 min following AAV or retrobead release. Coordinates were slightly adjusted based on mouse age and size.

For photometry fiber placement in the VTA or mPFC, mice were implanted with a 400 µm diameter 0.48 NA single mode optical fiber (Doric lenses, Quebec City, QC, Canada) directly above the VTA at –2.8 mm (AP); + 0.4 mm (ML); 4.3–4.5 mm (DV), mPFC +2.0 mm (AP); + 0.4 mm (ML); 1.3–1.6 mm (DV), 4–6 weeks after viral transduction. Behavioral experiments were conducted 7–12 days after implantation. Real-time photometry recording was performed during optical fiber implant above the VTA, for optimal targeting. When the fiber tip approached the VTA region, a continuous increase of fluorescence intensity was observed. The final position of implantation was determined by the cessation of further increases in fluorescence intensity. Mice recovered for at least 7 days after implantation.

## Behavior assays
### Aversive learning
P40-60 mice were used for behavioral assays with fiber photometry and chemogenetics experiments. The strong learned helplessness procedure consisted of two induction sessions (one session per day; 360 inescapable foot shocks per session; 0.3 mA, 3 s; random 1–15 s inter-shock intervals). Active/

Passive Avoidance Shuttle Boxes from MazeEngineers (Boston, MA) were used for the experiment. To assess the degree of aversive learning, test sessions (30 escapable foot shocks per session; 0.3 mA, 10 s; random 5–30 s inter-shock intervals) were conducted before induction, 24 hr after the last induction session, and following pharmacological manipulations. The testing was performed in a shuttle box (18 × 18 × 20 cm) equipped with a grid floor and a gate separating the two compartments. No conditioned stimulus was delivered either before or after the shocks. Escapes were scored when the animal shuttled between compartments during the shock. Escape latency was measured as the time from the start of the shock to the escape. The shock automatically terminated when the animal shuttled to the other compartment. Failures were scored when the animal failed to escape before the shock end. All behavioral assays were conducted during the active phase of the circadian cycle. Schematics with mice were made using BioRender.

To evaluate learned helplessness behavior in a novel context, another shuttle box with different decorative designs was used. In addition to a differently designed gate, decorations with black and white patterns (bars and circles) were applied to the walls. To evaluate the effect of ketamine on learned helplessness behavior, a single dose of ketamine (10 mg/kg b.w., i.p.) was given 48 hr after the last induction session, with the test session performed 4 hr later. For chemogenetic activation in mPFC, 48 hr after the last induction session, Clozapine N-oxide (CNO) was administered i.p., followed by test sessions 2 hr and 4 hr later. For chemogenetic inhibition in VTA, the first CNO dose (3 mg/kg, i.p.) was co-administered with ketamine 48 hr after the last induction, followed by test sessions 4 and 24 hr later. Then, immediately following the last test session ketamine was administered alone, followed by test sessions 4, 24, and 72 hr later.

For the weak aversive learning procedure, a single induction session (360 inescapable foot shocks; 0.3 mA, 3 s; random 1–15 s intershock intervals) was performed at the end of the second day. Six test sessions were performed (one session per day; 100 escapable foot shocks per session; 0.3 mA, 3 s; random 5–15 s intershock intervals). A single dose of ketamine (10 mg/kg b.w., i.p.) was given 4 hr before the fifth test session. The final test session was performed 24 hr after ketamine treatment. Age-matched controls were given six test sessions, in the absence of induction sessions or ketamine treatment.

### Locomotion test

To assess the locomotor activity, mice were placed in the center of a plastic chamber (48 cm × 48 cm × 40 cm) or in the shuttle box in a dimly lit room. Mice explored the arena for 15 min, with video (30 fps) and photometry recording performed during the final 10 min.

## Fiber photometry

Hardware was created based on open-source resources made available by Dr. Thomas Davidson (https://drive.google.com/drive/folders/0B7FioEJAlB1aNmdPOEsxTjhxajg) (*Lerner et al., 2015*). Briefly, a custom-built setup was created combining Doric fluorescence mini-cube (Doric, Westport, CT) and a 2151 Femtowatt photoreceiver with a lensed adapter (Newport, Irvine, CA). All downstream hardware including fiberoptic cannuale and patch cords, except for LEDs and drivers (Thorlabs, Newton, NJ), is readily available from Doric. A conventional single-cell electrophysiology recording system (DAQ+ software) was used to acquire signal and drive the LED, with a modified version of MATLAB based Scanimage (*Pologruto et al., 2003*) adapted for electrophysiology recordings by Dr. Bernardo Sabatini (https://github.com/bernardosabatinilab). Signals were sampled at 1 kHz and downsampled to 250 Hz for time-locked and to 10 Hz for non-time-locked analyses. Fluorescence signal was baseline adjusted in non-overlapping 100 sec windows as (signal-median(signal)) / median(signal), denoted as dF/F. Recordings were made during a subset of the sessions of LH and open-field locomotion, as noted in the text. Additional analyses details and link to analysis code are below.

## Acute slice preparation and electrophysiology

Coronal brain slice preparation was modified from previously published procedures (*Kozorovitskiy et al., 2015*; *Kozorovitskiy et al., 2012*; *Xiao et al., 2017*). Animals were deeply anesthetized by inhalation of isoflurane, followed by a transcardial perfusion with ice-cold, oxygenated artificial cerebrospinal fluid (ACSF) containing (in mM) 127 NaCl, 2.5 KCl, 25 NaHCO$_3$, 1.25 NaH$_2$PO$_4$, 2.0 CaCl$_2$, 1.0 MgCl$_2$, and 25 glucose (osmolarity 310 mOsm/L). After perfusion, the brain was rapidly removed,

and immersed in ice-cold ACSF equilibrated with 95%$O_2$/5%$CO_2$. Tissue was blocked and transferred to a slicing chamber containing ice-cold ACSF, supported by a small block of 4 % agar (Sigma-Aldrich). Bilateral 250 or 300 µm-thick slices were cut on a Leica VT1000s (Leica Biosystems, Buffalo Grove, IL) in a rostro-caudal direction and transferred into a holding chamber with ACSF, equilibrated with 95%$O_2$/5%$CO_2$. Slices were incubated at 34°C for 30 min prior to electrophysiological recording and two-photon calcium imaging. Slices were transferred to a recording chamber perfused with oxygenated ACSF at a flow rate of 2–4 ml/min at room temperature.

To assess spontaneous firing rate of dopaminergic neurons in the VTA, cell-attached recordings were performed. Cell-attached recording electrode pipettes were filled with the internal solution for voltage clamp recordings to monitor spontaneous break in, with pipette resistance varying between 3 and 7 MΩ. To assess spontaneous synaptic inputs (sEPSC/sIPSC), voltage clamp recordings were performed. Dopaminergic neurons were identified by the expression of mCherry or GFP in DAT$^{iCre}$ mice. Recording electrodes contained the following (in mM): Cell-attached recordings and voltage clamp for sEPSCs: 120 CsMeSO$_4$, 15 CsCl, 10 HEPES, 10 Na- phosphocreatine, 2 MgATP, 0.3 NaGTP, 10 QX314, and 1 EGTA (pH 7.2–7.3, ~ 295 mOsm/L); Voltage clamp for sIPSCs (high chloride internal solution): 100 CsCl, 35 CsF, 4 MgCl2, 10 HEPES, 10 Na-phosphocreatine, 4 MgATP, 0.4 Na2GTP, and 1 EGTA (pH 7.2, 295 mOsm/L). In voltage clamp recordings, cells were held at –70 mV. Recordings were made using 700B amplifiers (Axon Instruments, Union City, CA); data were sampled at 10 kHz and filtered at 4 kHz with a MATLAB-based acquisition script (MathWorks, Natick, MA). Offline analysis of electrophysiology data was performed using MATLAB (Mathworks, Natick, MA), and Clampfit 11.2 (Molecular Devices, San Jose, CA).

## Two-photon GCaMP6f imaging

Calcium sensor imaging was accomplished on a custom-built microscope combining two-photon laser-scanning microscopy (2PLSM), as previously described (*Banala et al., 2018*; *Kozorovitskiy et al., 2015*; *Kozorovitskiy et al., 2012*; *Xiao et al., 2018*; *Xiao et al., 2017*). A mode-locked Ti:Sapphire laser (Mai Tai eHP DeepSee, Spectra-Physics, Santa Clara, CA) was tuned to 910 nm for GCamp6f. The intensity of the laser was controlled by Pockels cells (Conoptics, Danbury, CT). A modified version of Scanimage software was used for data acquisition (*Pologruto et al., 2003*). Calcium imaging of GCaMP6f expressing DA neurons in acute brain slices of the VTA was done at 910 nm and sampled at 12 Hz. Spontaneous activity was imaged for 5 min in the baseline at 34°C, followed by a 5 min recording after ex vivo ketamine application (50 µM, with 15–40 min delay).

## Pharmacology

Pharmacological agents were acquired from Vedco (St. Joseph, MO) or Sigma-Aldrich (St. Louis, MO). In vivo injections included intraperitoneal and subcutaneous injections of ketamine (10 mg/kg, Vedco, St. Joseph, MO) and Clozapine N-oxide (3 mg/kg in vivo, 1 µM in vitro, Sigma-Aldrich); intracranial injections of ketamine (12.5 µg in 500 nl ACSF) and Chicago Sky Blue 6B (50 µg in 500 nl ACSF, Tocris, Bristol, United Kingdom). Ex vivo applications included SR95531 (Gabazine, 10 µM, Tocris), CNQX (10 µM, Tocris), and ketamine (50 µM).

## Tissue processing and immunohistochemistry

Mice were deeply anesthetized with isoflurane and transcardially perfused with 4% paraformaldehyde (PFA) in 0.1 M phosphate buffered saline (PBS). Brains were post-fixed for 1–5 days and washed in PBS, prior to sectioning at 50–100 µm on a vibratome (Leica Biosystems). Sections were pretreated in 0.2% Triton X-100 for an hour at RT, then blocked in 10% bovine serum albumin (BSA, Sigma-Aldrich, ST Louis, MO):PBS with 0.05 % Triton X-100 for 2 hr at RT, and incubated for 24–48 hr at 4°C with primary antibody solution in PBS with 0.2 % Triton X-100. On the following day, tissue was rinsed in PBS, reacted with secondary antibody for 2 hr at RT, rinsed again, then mounted onto Superfrost Plus slides (ThermoFisher Scientific, Waltham, MA). Sections were dried and coverslipped under ProLong Gold antifade reagent with DAPI (Molecular Probes, Life Technologies, Carlsbad, CA) or under glycerol:TBS (9:1) with Hoechst 33342 (2.5 µg/ml, ThermoFisher Scientific). Primary antibodies used in the study were rabbit anti-tyrosine hydroxylase (1:1000; AB152, Millipore, Burlington, MA), mouse anti-tyrosine hydroxylase (1:1000; AB129991, Abcam, Cambridge, UK), rabbit anti-RFP (1:500, 600-401-379, Rockland, Limerick, PA), and rabbit anti-c-Fos (1:5000; Synaptic Systems, Goettingen,

Germany). Alexa Fluor 488-, Fluor 594-, or Fluor 647-conjugated secondary antibodies against rabbit or mouse(Life Technologies, Carlsbad, CA) were diluted 1:500. Whole sections were imaged with an Olympus VS120 slide scanning microscope (Olympus Scientific Solutions Americas, Waltham, MA). Confocal images were acquired with a Leica SP5 confocal microscope (Leica Microsystems). Depth-matched z-stacks of 2 μm-thick optical sections were analyzed in ImageJ (FIJI) (*Schindelin et al., 2012*; *Schneider et al., 2012*). For c-fos quantification, every four adjacent z stack slices were combined, for a total of 6 μm thickness. mCherry signal was used to localize cell bodies of hM3Dq-expressing neurons. Laser intensity and all imaging parameters were held constant across samples, and the same threshold was applied for subtracting background immunofluorescence. C-fos+ neurons were identified by an experimenter blind to the conditions.

## Quantitative fluorescence in situ hybridization

Quantitative fluorescence in situ hybridization (FISH) was conducted following previously published procedures (*Xiao et al., 2018*; *Xiao et al., 2017*). Mice were deeply anesthetized by inhalation of isoflurane and decapitated. Brains were quickly removed and frozen in tissue-freezing medium on a mixture of dry ice and ethanol for 5–15 min prior to storage at 80°C. Brains were subsequently cut on a cryostat (Leica CM1850, Leica Biosystems) into 20-μm-thick sections, adhered to Superfrost Plus slides, and frozen at 80°C. Samples were fixed with 4% PFA in 0.1 M PBS at 4°C for 15 min, processed according to the manufacturer's instructions in the RNAscope Fluorescent Multiplex Assay manual for fresh frozen tissue (Advanced Cell Diagnostics, Newark, CA), and coverslipped with ProLong Gold antifade reagent with DAPI (Molecular Probes). *Drd1a* receptor channel 2 (*Drd1a*) or *Slc6a3* (*Dat*) probes were added to slides in combination, and Amp4-b fluorescent amplification reagent was used for all experiments. Sections were subsequently imaged on a Leica SP5 confocal microscope in four channels with a ×40 objective lens at a zoom of 1.4 and resolution of 512 × 512 pixels with 1.0 μm between adjacent z sections.

FISH images were analyzed using FIJI (*Schindelin et al., 2012*). Briefly, every four adjacent z stack slices were combined, for a total of 3 μm thickness, in order to minimize missed colocalization, while decreasing false positive colocalization driven by signal from cells at a different depth in a similar x-y position. All channels were thresholded. Cellular ROIs were defined using the retrobead+ channel information to localize cell bodies. FISH molecule puncta were counted within established cell boundaries. Whether a cell was considered positive for a given marker was determined by setting a transcript-dependent threshold of the number of puncta (e.g. over five puncta/soma for *Drd1a*+). These stringent parameters for co-localization and the challenges of quantifying low abundance receptor transcripts likely lead to underestimation of receptor-positive populations.

## Quantification of behavior

For evaluating locomotor behavior, Toxtrac (*Rodriguez et al., 2018*) was used to track the animal's position, defined by its body center position, and quantify the distance travelled in each session. To detect motion onset/offset, movement was defined by the body center moving >30 mm/s for at least 0.5 s. The associated $Ca^{2+}$ transients were then aligned to transitions between motion start and stop times and averaged across all animals. Video was recorded and the mouse was tracked with Toxtrac.

Three state transition models were constructed for a given condition and behavioral response, by counting the occurrence of each of the three possible 2-response sequences in all animals and then dividing by the total number of the given response. For example, an escape can be followed by another escape, a failure, or a premature trial. The probability of a failure following an escape was calculated by counting the number of failures that followed escapes and dividing it by the total number of escape trials. The graphs for the transition models were constructed using GraphViz (*Gansner and North, 2000*). The similarity between transition models was calculated as one minus the Frobenius norm (the Euclidean norm of a matrix) of the difference between models ($1 - norm\,(model_a - \,model_b)$). To depict individual learning trajectories, starting at zero, trajectory value incremented by one for each escape trial, decremented by one for each failure trial, and kept constant for spontaneous transitions in the shuttle box.

For the analysis of distributions of % failure to escape, the density histograms of percentage of failures were generated with a bin width = 4. Density histograms were fitted with the Gaussian distribution using fitdistrplus R package using a maximum likelihood estimation.

## Quantification of fiber photometry data

The heatmap of $Ca^{2+}$ transients was constructed by plotting single trial transients with signal amplitude depicted by color. The latency to peak was defined as the time from shock onset to the maximum $Ca^{2+}$ transient value within 8 s of shock onset. The negative and positive area under the curve (AUC) was calculated based on the direction of the peaks. Peaks were omitted from analyses if they were less than 5% of the distance from minimum to maximum dF/F or less than three times the standard deviation of the baseline period. The baseline was chosen by using the mean dF/F of the 5 s segment before the shock onset. For behavioral response-specific $Ca^{2+}$ transients, the transients for each behavioral response were averaged within a single subject, and then these data were averaged across all animals. The distance between escape and failure traces for each animal was computed as the Euclidean norm of the difference between the average of all their $Ca^{2+}$ transients on escape trials and the average of all their $Ca^{2+}$ transients on failure trials ($\|mean\ (escape\ trials) - mean\ (failure\ trials)\|_2$).

## Quantification of two-photon $Ca^{2+}$ sensor imaging

ROIs of dopaminergic neuron somata were defined manually. Raw fluorescence intensity for all frames during 5-min recording sessions was extracted using FIJI (*Schindelin et al., 2012*). For each neuron, the power spectral density of its baseline adjusted fluorescence was computed using Welch's method (*Welch, 1967*) with a Hann window and 50% overlap.

## Statistical analyses

Required sample sizes were estimated based on previous publications and past experience. The number of replicates was reported, and several internal replications are present in the study. No data were excluded after analyses. Animals were randomly assigned to treatment groups. Group statistical analyses were done using GraphPad Prism 7 and 8 software (GraphPad, LaJolla, CA). For N sizes, the number of trials or cells recorded, as well as the number of animals are provided. All data are expressed as mean ± SEM, or individual plots. Probabilities are expressed as aggregate probabilities within individuals. For two-group comparisons, statistical significance was determined by two-tailed Student's t-tests. For multiple group comparisons, one-way or two-way analysis of variance (ANOVA) tests were used for normally distributed data, followed by post hoc analyses. For non-normally distributed data, non-parametric tests for the appropriate group numbers were used. Pearson regression was used to detect the correlation between two groups of data. $p < 0.05$ was considered statistically significant. All computer code generated in the current study is available at https://github.com/KozorovitskiyLaboratory/Wu_et_al_2021 (*Wu et al., 2021b*; copy archived at swh:1:rev:459ff15a97a2af4e955d20531235b399b4be5b22).

# Acknowledgements

The authors are grateful to Lindsey Butler for mouse colony management, Northwestern Biological Imaging Facility and Dr. Tiffany Schmidt for confocal microscope access. This work was supported by NINDS R01NS107539, NIMH R01MH117111, Rita Allen Foundation Scholar Award, Searle Scholar Award, Beckman Young Investigator Award, William and Bernice E Bumpus Young Innovator Award, NARSAD Young Investigator, and P&S Fund Grant (all YK). MW was supported as an affiliate fellow of the NIH T32 AG20506, SM is a fellow of the National Science Foundation Graduate Research Fellowship DGE-1842165, and VD is a predoctoral fellow of the American Heart Association (19PRE34380056).

# Additional information

## Funding

| Funder | Grant reference number | Author |
|---|---|---|
| Rita Allen Foundation | Rita Allen Foundation Scholar Award | Yevgenia Kozorovitskiy |
| National Institutes of Health | R01NS107539 | Yevgenia Kozorovitskiy |

| Funder | Grant reference number | Author |
|---|---|---|
| Kinship Foundation | Searle Scholar Award | Yevgenia Kozorovitskiy |
| Arnold and Mabel Beckman Foundation | Beckman Young Investigator Award | Yevgenia Kozorovitskiy |
| Brain and Behavior Research Foundation | NARSAD and P&S Fund Award | Yevgenia Kozorovitskiy |
| National Institutes of Health | T32 AG20506 affiliate fellow | Mingzheng Wu |
| National Science Foundation | GRFP DGE-1842165 | Samuel Minkowicz |
| American Heart Association | 19PRE34380056 predoctoral fellowship | Vasin Dumrongprechachan |
| National Institutes of Health | R01MH117111 | Yevgenia Kozorovitskiy |

The funders had no role in study design, data collection and interpretation, or the decision to submit the work for publication.

#### Author contributions

Mingzheng Wu, Conceptualization, Data curation, Formal analysis, Investigation, Methodology, Resources, Software, Validation, Visualization, Writing – original draft, Writing – review and editing; Samuel Minkowicz, Data curation, Formal analysis, Investigation, Methodology, Resources, Software, Visualization, Writing – original draft, Writing – review and editing; Vasin Dumrongprechachan, Investigation, Visualization, Writing – review and editing; Pauline Hamilton, Lei Xiao, Investigation, Writing – review and editing; Yevgenia Kozorovitskiy, Conceptualization, Data curation, funding-acquisition, Investigation, Methodology, project-administration, Resources, Software, supervision, Validation, Visualization, Writing – original draft, Writing – review and editing

#### Author ORCIDs

Mingzheng Wu (iD) http://orcid.org/0000-0003-4415-6296
Samuel Minkowicz (iD) http://orcid.org/0000-0003-1555-1158
Vasin Dumrongprechachan (iD) http://orcid.org/0000-0001-5890-6778
Lei Xiao (iD) http://orcid.org/0000-0002-1640-9690
Yevgenia Kozorovitskiy (iD) http://orcid.org/0000-0002-3710-1484

#### Ethics

Animals were handled according to protocols approved by the Northwestern University Animal Care and Use Committee (IS00000707).

#### Decision letter and Author response

Decision letter https://doi.org/10.7554/eLife.64041.sa1
Author response https://doi.org/10.7554/eLife.64041.sa2

## Additional files

#### Supplementary files
• Transparent reporting form

#### Data availability

All data generated or analyzed during this study are included in the manuscript and supporting files. Source data files are provided for each figure.

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
