## [Decision Letter]

**Acceptance summary:**

The authors employed fiber photometry-based in vivo calcium imaging, pharmacological manipulations, and the learned helplessness behavioral model in mice to show that aversive states and helpless behavior are associated with attenuated dopaminergic transmission, specifically from dopaminergic neurons of the ventral tegmental area. Importantly, the authors demonstrate that ketamine can restore both the dopaminergic and behavioral deficits associated with learned helplessness. This is a timely observation, considering the broad interest in understanding how ketamine - which was recently approved by the FDA as a treatment of depression - yields its known behavioral effects.

**Decision letter after peer review:**

Thank you for submitting your article "Attenuated dopamine signaling after aversive learning is restored by ketamine to rescue escape actions" for consideration by *eLife*. Your article has been reviewed by 3 peer reviewers, and the evaluation has been overseen by a Reviewing Editor and Kate Wassum as the Senior Editor. The following individual involved in review of your submission has agreed to reveal their identity: Manuel Mameli (Reviewer #3).

The reviewers have discussed the reviews with one another and the Reviewing Editor has drafted this decision to help you prepare a revised submission.

Summary:

The authors employed fiber photometry-based in vivo calcium imaging, pharmacological manipulations, and the learned helplessness behavioral model in mice to show that aversive states and helpless behavior are associated with attenuated dopaminergic transmission, specifically from dopaminergic neurons of the ventral tegmental area. Importantly, the authors demonstrate that ketamine can restore both the dopaminergic and behavioral deficits associated with learned helplessness. This is a timely observation, considering the broad interest in understanding how ketamine – which was recently approved by the FDA as a treatment of depression – yields its known behavioral effects.

Essential revisions:

The reviewers were collectively enthusiastic about the study, highlighting that the work is elegant and that key findings are robust. However, they agree that before a final decision can be made on the manuscript, a series of important issues must be addressed. As you will see, while for many of the points raised by the Reviewers additional discussion or highlighting of caveats will suffice, in some case additional experimental work is required. Each of the Reviewers' points is described in detail below.

1. The authors claim that ketamine rescues deficits in escape behavior induced by LH via VTA DA neuron activation mediated by a recurrent VTA – mPFC D1 neuron circuit. The data supporting a role for VTA DA neurons in ketamine's effect on escape behavior are convincing, however, the recurrent circuit component of the study is lacking a causal link. The authors fail to reject the recurrent circuit hypothesis based on (1) the lack of an effect of ketamine on VTA DA neuron GCaMP responses in brain slices, (2) the presence of Drd1a mRNA in the majority of VTA-projecting PFC neurons, (3) an increase in spontaneous GCaMP6 fluorescence in mPFC D1 neurons acutely and 24 hr after ketamine, (4) rescue of VTA DA dynamics during inescapable shocks by intra-mPFC ketamine, and (5) a rescue of escape responses after chemogenetic activation of global mPFC D1 neurons. All above observations, with the exception of the retrograde tracing/in-situ hybridization experiment, are agnostic to the projection target of VTA or mPFC neurons. Therefore, they do not differentiate between the recurrent hypothesis proposed by the authors and alternative hypotheses, such as activation of DA neurons that project to the nucleus accumbens or amygdala and control avoidance behavior (Darvas et al. 2020 Learning and Memory; Wenzel et al. 2018 Current Biology; Stelly et al. 2020 PNAS; among many others) or activation of PFC D1 neurons that project to the BLA shown to mediate antidepressant effects and be involved in ketamine's antidepressant effects (Hare et al. 2019 Nat Comm). The authors should discuss the assumptions/caveats associated with their present findings or provide additional experiments aimed at supporting this conclusion.

2. Related to the previous point. The mPFC projection is among the sources of glutamate to VTA DA neurons, yet not the only one; importantly not the only one sensitive to ketamine. The authors state the importance of mPFC but cannot rule out whether or not ketamine is also acting elsewhere. The authors should present additional experimental evidence demonstrating selectivity for the mPFC.

3. Did the authors conduct fiber photometry recordings from mice treated with saline for the wLH? The authors did include this control group in the behavioral studies in figure 1 but the lack of this in photometry experiments should be addressed as animals must adapt behavior in light of changing contingencies during behavioral training. In the wLH procedures the animal goes from a setting where escape behavior is reinforced (baseline days 1 and 2) to a setting where there is extinction of the escape response (wLH after day 2), and subsequent reinstatement of the avoidance behavior with reinforcement of escape behavior (days 3-6). Saline controls will be necessary to determine how much of the ketamine rescue effect is contributed by on-going reinforcement behavior after the wLH induction.

4. Were the mice that were used for the photometry recordings performed during inescapable footshocks testing (strong LH) in figure 1 also exposed to baseline escapable shocks? Similar to the previous comment, the history of reinforcement of escape behavior is predicted to shape responses during inescapable shocks. The authors should clarify this in the manuscript.

5. For the intra-mPFC ketamine rescue in figure 4 were the same mice plotted in figure 4g also used to generate data in figure 4h? Were responses during the baseline escapable shocks recorded? The authors should also provide recordings obtained from their control mice.

6. For the wLH experiments in figure 2, the mice in the wLH group are already showing increased failures, despite the text stating the wLH induction occurred at the end of day 2. This would suggest that mice in this group were already developing higher levels of failed avoidance from the escapable footshock testing relative to controls undergoing the same procedures. This should be clarified/addressed. Were the behavioral data in figure 2B from both controls and wLH mice separate that underwent photometry recordings? Were any recordings performed in control animals?

7. The authors conclude that ketamine has no effect on VTA DA neurons based on slice GCaMP imaging. However, NMDA receptors, a major target of ketamine, are unlikely to regulate tonic firing under these conditions but may be important for regulating plasticity of inputs and bursting in-vivo or in slices when afferent inputs might be engaged (Wang et al. 2011; Zweiffel et al. 2008; Jastrzebska et al. 2016; Mansvelder and McGehee 2000 Neuron; Yuan et al. 2013 Neuron). The authors should use other methods besides calcium imaging in slice to examine the effect of ketamine ex vivo. Electrophysiological studies will be more sensitive and reveal additional local effects, and should be conducted to control for effects directly…and sensitivity of calcium imaging dynamics.

8. The authors should also record and plot the failure rate in trials alongside the DRD1 neuronal activity following LH induction? They may already have this data, and it would be useful to see how that fits with their model.

9. A group of mice whereby they authors use a chemogenetic approach or optogenetic method to inhibit/silence the Drd1 neurons in the mPFC during ketamine exposure in the LH induced mice, to see if this manipulation blocks the rescue of the escape behavior.

10. Recent data suggest that the action of Ketamine in restoring depressive-like symptoms, likely including escapable actions, relies on targeting a variety of brain structures like the mPFC (Liston Laboratory) or the astrocytes function in the lateral habenula (Hu Laboratory). The authors should discuss this mechanistic debate around ketamine's action.

11. The field largely discusses how low doses of ketamine can be efficient in restoring behavioral adaptations. Do the authors have a dose response for their behavioral and functional data? If so, it would be good to include this information in the manuscript, as it may boost the clinical importance of such study.

12. Dopamine neurons subpopulations underlie discrete behaviors. Do the authors think their dopamine responses are homogeneous throughout the VTA, or in light of the circuits they are embedded? This should be either tested experimentally or at least discussed.

13. The authors performed an analysis to obtain a relationship between an animal's ketamine responsiveness and VTA DA activity. Several laboratories reported that the effectiveness of the LH protocol only accounts for a fraction of animals. In other words a fraction of animals is resilient. How is DA activity in these animals? Does VTA DA activity represents a "biomarker" for susceptibility versus resilience? The authors could use existing behavioral data to address this point.

---

## [Author Response]

Essential revisions:

The reviewers were collectively enthusiastic about the study, highlighting that the work is elegant and that key findings are robust. However, they agree that before a final decision can be made on the manuscript, a series of important issues must be addressed. As you will see, while for many of the points raised by the Reviewers additional discussion or highlighting of caveats will suffice, in some case additional experimental work is required. Each of the Reviewers' points is described in detail below.

1. The authors claim that ketamine rescues deficits in escape behavior induced by LH via VTA DA neuron activation mediated by a recurrent VTA – mPFC D1 neuron circuit. The data supporting a role for VTA DA neurons in ketamine's effect on escape behavior are convincing, however, the recurrent circuit component of the study is lacking a causal link.

We agree with the Reviewer’s that while the significance of the actions of VTA DA neurons is well supported by this manuscript, we have not provided sufficient evidence directly supporting the importance of recurrent connectivity. We have removed claims regarding causal involvement of the VTA-mPFC recurrent circuit in the regulation of escape actions, reviewing a number of key other circuits that are likely involved. Instead, we propose potential future directions of study related to this hypothetical framework in the Discussion section. Lines: 378-387, 398-432.

The authors fail to reject the recurrent circuit hypothesis based on 1) the lack of an effect of ketamine on VTA DA neuron GCaMP responses in brain slices, 2) the presence of Drd1a mRNA in the majority of VTA-projecting PFC neurons, 3) an increase in spontaneous GCaMP6 fluorescence in mPFC D1 neurons acutely and 24 hr after ketamine, 4) rescue of VTA DA dynamics during inescapable shocks by intra-mPFC ketamine, and 5) a rescue of escape responses after chemogenetic activation of global mPFC D1 neurons. All above observations, with the exception of the retrograde tracing/in-situ hybridization experiment, are agnostic to the projection target of VTA or mPFC neurons. Therefore, they do not differentiate between the recurrent hypothesis proposed by the authors and alternative hypotheses, such as activation of DA neurons that project to the nucleus accumbens or amygdala and control avoidance behavior (Darvas et al. 2020 Learning and Memory; Wenzel et al. 2018 Current Biology; Stelly et al. 2020 PNAS; among many others) or activation of PFC D1 neurons that project to the BLA shown to mediate antidepressant effects and be involved in ketamine's antidepressant effects (Hare et al. 2019 Nat Comm). The authors should discuss the assumptions/caveats associated with their present findings or provide additional experiments aimed at supporting this conclusion.

We agree with the Reviewers that DA signaling in many brain regions other than mPFC is involved in the regulation of escape actions and avoidance behaviors. For example, the release and postsynaptic function of DA in the NAc and amygdala have been extensively studied in conditioned avoidance learning (Antunes et al., 2020; Darvas et al., 2011; McCullough et al., 1993; Oleson et al., 2012; Oleson and Cheer, 2013; Stelly et al., 2019; Wenzel et al., 2018). In our study, we found that VTA DA activity is not necessarily critical for innate escape actions, since chemogenetic suppression of VTA DA activity does not increase failures to escape from shock stimuli in the absence of prior learning. Importantly, after LH, responses of VTA DA neurons to both inescapable and escapable shocks are significantly changed. Since DA release has been shown to be highly correlated with somatic DA neuronal activity (de Jong et al., 2019; Lee et al., 2020; Patriarchi et al., 2018), we predict that downstream DA release in the NAc and amygdala are also modulated after learned helplessness, and after ketamine treatment. Given the importance of NAc and amygdala DA signaling in avoidance learning and maintenance (Antunes et al., 2020; Darvas et al., 2011; Stelly et al., 2019), behavioral outcomes to escapable shock may depend on DA release across multiple regions. We have expanded relevant components in the Discussion section. Lines: 378-387.

DA signaling in the mPFC regulates the processing of aversive stimuli, food intake, and other behaviors (Land et al., 2014; Shinohara et al., 2018; Vander Weele et al., 2018; Wass et al., 2018; Winter et al., 2009). A recent publication from Hare et al. showed that activation of mPFC D1 neurons projections in the BLA produced antidepressant-like effects (Hare et al., 2019). Importantly, BLA receives DA inputs from the VTA and is involved in avoidance learning. Therefore, after ketamine treatment, BLA may serve as a convergence site that synthesizes inputs from VTA and mPFC to facilitate the rescue of escape actions.

Thus, we do not suggest that the mPFC and VTA are the only brain regions that control escape behaviors. However, we think the enhanced activity within these two brain regions is one of the important mechanisms underlying behavioral effects of ketamine. Our data show that:

1. Local mPFC ketamine infusion rescues VTA DA activity and behavior (Figure 5b, d)

*2.* In vivo ketamine treatment rapidly enhances D1 neuronal activity in mPFC (Figure 6—figure supplement 1c, d)

3. Chemogenetic activation of mPFC D1 neurons rescues escape behaviors (Figure 6g)

Moreover, in recently published work from our lab, we show that DA signaling through Drd1 activation is necessary for ketamine’s action on restoring glutamate-dependent dendritic spinogenesis (Wu et al., 2021), which has been shown to maintain the lasting behavioral effects of ketamine after corticosterone treatment (Moda-Sava et al., 2019). Chemogenetic activation of Gα_s_-coupled signaling cascades downstream of Drd1 activation in mPFC also rescues escape after LH as well (Wu et al., 2021). Based on the sum of these data, we hypothesize that the initial enhancement of activity induced by ketamine in mPFC may be amplified within this brain circuit, which may potentially extend ketamine’s behavioral effects from hours to days. This hypothesis, along with important caveats and alternative interpretations, is presented in the Discussion section. Lines: 398-432.

2. Related to the previous point. The mPFC projection is among the sources of glutamate to VTA DA neurons, yet not the only one; importantly not the only one sensitive to ketamine. The authors state the importance of mPFC but cannot rule out whether or not ketamine is also acting elsewhere. The authors should present additional experimental evidence demonstrating selectivity for the mPFC.

In addition to our prior experiments that directly infuse ketamine into mPFC, we have now added new experiments where we infused ketamine locally into another three brain regions (PAG, PPTg, and VTA) followed by behavioral testing. We selected PAG and PPTg for two reasons: (1) these brain regions contain a large number of glutamatergic neurons that regulate the activity of DA neurons in the VTA (Morales and Margolis, 2017); (2) behavioral effects of local ketamine infusion into these regions have not been previously demonstrated, to our knowledge, although antidepressant effects of local PAG infusion of (2R,6R)-HNK have been reported (Chou et al., 2018). We also carried out local infusions of ketamine into the VTA itself, to evaluate the possibility of direct in vivo effects there (in addition to our prior and new data showing a lack of effects of ketamine on VTA DA neuron synaptic inputs and activity in the acute slice preparation, Figure 4). Our results show that local infusion of ketamine (12.5 µg in 500 nl ACSF) into any selected region other than mPFC does not rescue escape behavior after LH, supporting the idea that not all glutamatergic inputs into the VTA are modulated by ketamine to rescue escape actions (Figure 5b). Lines: 259-269.

We agree with the Reviewers that the mPFC is not the only region sensitive to ketamine. It is very likely that ketamine exerts antidepressant effects through synergistic effects across many brain regions, including direct effects and indirect consequences of changes in cortical neural activity. For example, prior studies have demonstrated that local infusion of ketamine into DG/CA3 of hippocampus increases escape behaviors after LH (Shirayama and Hashimoto, 2017) and local infusions of ketamine into lateral habenula improves depressive-like behaviors (Y. Yang et al., 2018). Nevertheless, our data suggest that the disinhibition and activation of mPFC neurons, as shown in this study (for Drd1 neurons) and other studies (for broader populations, e.g., (Hare et al., 2020)), may serve as an important mechanism by which ketamine recovers the activity of VTA DA neurons. Lines: 423-432.

3. Did the authors conduct fiber photometry recordings from mice treated with saline for the wLH? The authors did include this control group in the behavioral studies in figure 1 but the lack of this in photometry experiments should be addressed as animals must adapt behavior in light of changing contingencies during behavioral training. In the wLH procedures the animal goes from a setting where escape behavior is reinforced (baseline days 1 and 2) to a setting where there is extinction of the escape response (wLH after day 2), and subsequent reinstatement of the avoidance behavior with reinforcement of escape behavior (days 3-6). Saline controls will be necessary to determine how much of the ketamine rescue effect is contributed by on-going reinforcement behavior after the wLH induction.

In new experiments, we carried out saline injections on Day 5 after wLH and recorded the activity of VTA DA neurons during escapable shocks. Saline injections do not reduce failures to escape on Days 5-6 relative to Days 3-4. A single dose of ketamine treatment suffices to reduce failures compared to saline-treated animals (Figure 2b). Also, saline treatment does not change the activity of DA neurons during either escape or failure trials (Figure 2—figure supplement 4a, b). Together, our data suggest that reduced failures and restored DA dynamics on Day 5-6 after ketamine treatment is unlikely to be driven by ongoing reinforcement behavior after wLH induction. Lines: 188-190.

4. Were the mice that were used for the photometry recordings performed during inescapable footshocks testing (strong LH) in figure 1 also exposed to baseline escapable shocks? Similar to the previous comment, the history of reinforcement of escape behavior is predicted to shape responses during inescapable shocks. The authors should clarify this in the manuscript.

We have clarified that the animals were exposed to escapable foot shocks prior to LH induction. Line: 107.

5. For the intra-mPFC ketamine rescue in figure 4 were the same mice plotted in figure 4g also used to generate data in figure 4h? Were responses during the baseline escapable shocks recorded? The authors should also provide recordings obtained from their control mice.

The responses during baseline escapable shocks were not recorded in the initial experiments. We carried out new photometry recording in a separate group of mice receiving local ACSF infusions in mPFC. Our results show that local ACSF infusion in mPFC does not significantly change the responses of VTA DA to inescapable shocks after LH, providing an important control experiment (Figure 5—figure supplement 1b, c). Lines: 273-274.

6. For the wLH experiments in figure 2, the mice in the wLH group are already showing increased failures, despite the text stating the wLH induction occurred at the end of day 2. This would suggest that mice in this group were already developing higher levels of failed avoidance from the escapable footshock testing relative to controls undergoing the same procedures. This should be clarified/addressed. Were the behavioral data in figure 2B from both controls and wLH mice separate that underwent photometry recordings? Were any recordings performed in control animals?

The Reviewers noted a small difference across days 1 and 2 in baseline performance of animals prior to LH induction; however, this difference did not reach statistical significance on a two-way ANOVA post hoc comparison:

We included new behavioral data in the control group for obtaining photometry recordings discussed in the paragraph below (Figure 2—figure supplement 3) and carried out two-way ANOVA analysis in panel 2b to compare escape behaviors across different days and between groups (control vs wLH). Both row factor (days) and column factor (groups) significantly contribute to the variation (p < 0.0001). We performed multiple comparisons to specifically determine the difference across days. We did not observe a significant difference between Day 1 and Day 2 (Sidak’s test, Day 1 vs Day 2, p = 0.1862), among the two groups on Day 1 (Sidak’s test, wLH vs control, p = 0.9934), or on Day 2 (Sidak’s test, wLH vs control, p = 0.1147). We do see a non-significant trend towards increased failures in the wLH group at Day 2 relative to Day 1 (Sidak’s test, p = 0.1010) and we agree that some form of experience-dependent learning occurs when animals are repeatedly exposed to escapable shocks. However, the wLH induction with inescapable shocks is necessary to induce aversive learning as operationalized in this study, since escapable shocks alone in the control group do not change daily escape actions (Figure 2b). Lines: 160-161.

Further, we carried out photometry recording in a separate group of control mice that did not go through wLH induction but were only exposed to escapable shocks. We compared the activity patterns of VTA DA in escape and failure trials across days. Responses of VTA DA during escapes and during failures did not change across days (Figure 2—figure supplement 3a, b). These data support the argument that wLH induction with inescapable shocks is necessary to reshape VTA DA responses. Lines: 186-188.

7. The authors conclude that ketamine has no effect on VTA DA neurons based on slice GCaMP imaging. However, NMDA receptors, a major target of ketamine, are unlikely to regulate tonic firing under these conditions but may be important for regulating plasticity of inputs and bursting in-vivo or in slices when afferent inputs might be engaged (Wang et al. 2011; Zweiffel et al. 2008; Jastrzebska et al. 2016; Mansvelder and McGehee 2000 Neuron; Yuan et al. 2013 Neuron). The authors should use other methods besides calcium imaging in slice to examine the effect of ketamine ex vivo. Electrophysiological studies will be more sensitive and reveal additional local effects, and should be conducted to control for effects directly…and sensitivity of calcium imaging dynamics.

We have carried out new cell-attached electrophysiological recordings and whole-cell voltage clamp recordings to further investigate the potential direct local effects of ketamine on genetically targeted VTA DA neurons (Figure 4g-o). Consistent with our prior GCaMP imaging results, ex vivo application of ketamine did not change the spontaneous firing rate of VTA DA cells that we recorded in cell-attached mode (Figure 4g-i). Furthermore, our results from whole-cell voltage clamp recordings show that ex vivo application of ketamine does not significantly change the amplitudes or inter-event intervals of spontaneous excitatory or inhibitory postsynaptic currents, EPSCs and IPSCs, recorded in separate experiments (Figure 4j-o). In addition, in vivo local injection of ketamine directly into the VTA does not rescue escape actions after LH (Figure 5b). Although glutamatergic receptors on VTA DA neurons are of course critical for many aspects of their function (Engblom et al., 2008; Sun et al., 2005; Wang et al., 2011; Zweifel et al., 2008), our data suggest the in vivo effects of ketamine on DA activity and behavior are not likely to be caused by local ketamine action in the VTA. Lines: 251-257.

8. The authors should also record and plot the failure rate in trials alongside the DRD1 neuronal activity following LH induction? They may already have this data, and it would be useful to see how that fits with their model.

We did not carry out behavioral manipulations in this experiment. The goal of the experiment was to simply evaluate baseline ketamine effects on the Drd1 neuronal activity in mPFC. The increased activity we observed in Drd1 neurons is consistent with a prior study that demonstrated a rapid enhancement of neuronal activity in mPFC pyramidal neurons following ketamine treatment, where mPFC neurons were broadly targeted via a CamKII promoter for GCaMP expression (Hare et al., 2020).

9. A group of mice whereby they authors use a chemogenetic approach or optogenetic method to inhibit/silence the Drd1 neurons in the mPFC during ketamine exposure in the LH induced mice, to see if this manipulation blocks the rescue of the escape behavior.

We carried out new experiments using chemogenetic approaches to inhibit mPFC Drd1 neurons following ketamine treatment after LH. Our results showed that inhibition of mPFC Drd1^+^ neuronal excitability blocked the behavioral effects of ketamine (Figure 6—figure supplement 2a, b), consistent with a recently published study (Hare et al., 2019). Altogether, these data confirm that the activation of mPFC Drd1 neurons is important for the behavioral effects of ketamine. Lines: 306-308.

10. Recent data suggest that the action of Ketamine in restoring depressive-like symptoms, likely including escapable actions, relies on targeting a variety of brain structures like the mPFC (Liston Laboratory) or the astrocytes function in the lateral habenula (Hu Laboratory). The authors should discuss this mechanistic debate around ketamine's action.

Given the lack of ketamine’s local, direct effects on VTA DA neuronal activity, the modulation of VTA DA activity by ketamine *in vivo* must depend on changes in the activity of projections from other brain regions. Numerous studies demonstrate that mPFC is broadly involved in the rapid and lasting behavioral effects of ketamine (Homayoun and Moghaddam, 2007; Kinoshita et al., 2018; Li et al., 2010; Lindefors et al., 1997; Liu et al., 2013; Lorrain et al., 2003; Moda-Sava et al., 2019; Phoumthipphavong et al., 2016; Wu et al., 2021). Another such potential mechanism could involve actions in the lateral habenula, which sends glutamatergic inputs to GABAergic neurons that suppress VTA DA activity (Gonçalves et al., 2012). We have expanded the Discussion section to ground the new findings in the literature on the functions of mPFC and lateral habenula in ketamine’s effects. Lines: 363-377, 426-432.

11. The field largely discusses how low doses of ketamine can be efficient in restoring behavioral adaptations. Do the authors have a dose response for their behavioral and functional data? If so, it would be good to include this information in the manuscript, as it may boost the clinical importance of such study.

Dose responses to ketamine have been carefully mapped out in human patients and animal models (Berman et al., 2000; Daly et al., 2018; Glue et al., 2011; Kurdi et al., 2014; Laskowski et al., 2011; Sassano-Higgins et al., 2016; Zarate et al., 2006). We do not have a dose response in our behavioral and functional data, because this detailed knowledge is available in the field (Author response image 1). The chosen sub-anesthetic dose of ketamine should approximately match the human clinically relevant dosing (Author response image 1, Nair and Jacob, 2016) and matches the dosing used in other animal studies (Ali et al., 2020; Belujon and Grace, 2014; Li et al., 2010; Miller et al., 2014; Moda-Sava et al., 2019).

**Author response image 1. sa2fig1:** (**a**) Ketamine dosage in clinical practice. Figure adapted from Glue et al., 2011. (**b**). Dose conversion calculation between rodents and humans. Table adapted from Nair and Jacob, 2016.

12. Dopamine neurons subpopulations underlie discrete behaviors. Do the authors think their dopamine responses are homogeneous throughout the VTA, or in light of the circuits they are embedded? This should be either tested experimentally or at least discussed.

VTA DA neurons are highly heterogenous based on their input/output anatomy and transcriptional profiles (Beier et al., 2015; Do et al., 2016; Morales and Margolis, 2017). Prior studies have demonstrated that VTA DA neurons respond to foot shocks differently, based on the brain regions they project to de Jong et al., 2019. In this study, we do not distinguish responses of DA neurons based on their projections or transcriptional profiles. We consider this a strength. The observed effects are sufficiently powerful to be seen on the background of projection target diversity. We have expanded the Discussion section to consider important points related to VTA DA neuron heterogeneity. Lines: 388-396.

13. The authors performed an analysis to obtain a relationship between an animal's ketamine responsiveness and VTA DA activity. Several laboratories reported that the effectiveness of the LH protocol only accounts for a fraction of animals. In other words a fraction of animals is resilient. How is DA activity in these animals? Does VTA DA activity represents a "biomarker" for susceptibility versus resilience? The authors could use existing behavioral data to address this point.

In this study, the distribution of behavioral outcomes after LH does not appear to be bimodal (Figure 6—figure supplement 3a, b), so we elected to not force a binary classification but display the full distributions of data. Lines: 341-344.

We observed biologically significant variability in VTA DA response which correlated with future behavioral outcomes. When VTA DA neuron activity substantially distinguishes failures to escape from successful escapes, individuals show less failures after LH. Based on this observation, it is expected that task specific VTA DA activity may represent the degree of susceptibility to stressful events as a potential biomarker. In the context of human populations, PET studies have been used to measure aspects of DA system function (Jönsson et al., 1999; Volkow et al., 1996). Importantly, human genetic variability, which can drive clinically relevant differences in DA signaling (Jönsson et al., 1999), could provide additional biomarkers to predict ketamine efficacy. *Lines: 338-354.*

References:

Ali F, Gerhard DM, Sweasy K, Pothula S, Pittenger C, Duman RS, Kwan AC. 2020. Ketamine disinhibits dendrites and enhances calcium signals in prefrontal dendritic spines. Nat Commun 11:1–15. doi:10.1038/s41467-019-13809-8

Antunes GF, Gouveia FV, Rezende FS, Seno MD de J, de Carvalho MC, de Oliveira CC, dos Santos LCT, de Castro MC, Kuroki MA, Teixeira MJ, Otoch JP, Brandao ML, Fonoff ET, Martinez RCR. 2020. Dopamine modulates individual differences in avoidance behavior: A pharmacological, immunohistochemical, neurochemical and volumetric investigation. Neurobiol Stress 12:100219. doi:10.1016/j.ynstr.2020.100219

Bagot RC, Cates HM, Purushothaman I, Vialou V, Heller EA, Yieh L, LaBonté B, Peña CJ, Shen L, Wittenberg GM, Nestler EJ. 2017. Ketamine and Imipramine Reverse Transcriptional Signatures of Susceptibility and Induce Resilience-Specific Gene Expression Profiles. Biol Psychiatry 81:285–295. doi:10.1016/j.biopsych.2016.06.012

Bagot RC, Parise EM, Peña CJ, Zhang HX, Maze I, Chaudhury D, Persaud B, Cachope R, Bolaños-Guzmán CA, Cheer J, Deisseroth K, Han MH, Nestler EJ. 2015. Ventral hippocampal afferents to the nucleus accumbens regulate susceptibility to depression. Nat Commun 6:1–9. doi:10.1038/ncomms8062

Beier KT, Steinberg EE, DeLoach KE, Xie S, Miyamichi K, Schwarz L, Gao XJ, Kremer EJ, Malenka RC, Luo L. 2015. Circuit Architecture of VTA Dopamine Neurons Revealed by Systematic Input-Output Mapping. Cell 162:622–634. doi:10.1016/J.CELL.2015.07.015

Belujon P, Grace AA. 2014. Restoring mood balance in depression: Ketamine reverses deficit in dopamine-dependent synaptic plasticity. Biol Psychiatry 76:927–936. doi:10.1016/j.biopsych.2014.04.014

Berman RM, Cappiello A, Anand A, Oren DA, Heninger GR, Charney DS, Krystal JH. 2000. Antidepressant effects of ketamine in depressed patients. Biol Psychiatry 47:351–354. doi:10.1016/S0006-3223(99)00230-9

Chou D, Peng HY, Lin T Bin, Lai CY, Hsieh MC, Wen YC, Lee AS, Wang HH, Yang PS, Chen G Den, Ho YC. 2018. (2R,6R)-hydroxynorketamine rescues chronic stress-induced depression-like behavior through its actions in the midbrain periaqueductal gray. Neuropharmacology 139:1–12. doi:10.1016/j.neuropharm.2018.06.033

Covington HE, Lobo MK, Maze I, Vialou V, Hyman JM, Zaman S, LaPlant Q, Mouzon E, Ghose S, Tamminga CA, Neve RL, Deisseroth K, Nestler EJ. 2010. Antidepressant effect of optogenetic stimulation of the medial prefrontal cortex. J Neurosci 30:16082–16090. doi:10.1523/JNEUROSCI.1731-10.2010

Daly EJ, Singh JB, Fedgchin M, Cooper K, Lim P, Shelton RC, Thase ME, Winokur A, Van Nueten L, Manji H, Drevets WC. 2018. Efficacy and safety of intranasal esketamine adjunctive to oral antidepressant therapy in treatment-resistant depression: A randomized clinical trial. JAMA Psychiatry 75:139–148. doi:10.1001/jamapsychiatry.2017.3739

Darvas M, Fadok JP, Palmiter RD. 2011. Requirement of dopamine signaling in the amygdala and striatum for learning and maintenance of a conditioned avoidance response. Learn Mem 18:136–143. doi:10.1101/lm.2041211

de Jong JW, Afjei SA, Pollak Dorocic I, Peck JR, Liu C, Kim CK, Tian L, Deisseroth K, Lammel S. 2019. A Neural Circuit Mechanism for Encoding Aversive Stimuli in the Mesolimbic Dopamine System. Neuron 101:133-151.e7. doi:10.1016/J.NEURON.2018.11.005

Do JP, Xu M, Lee S-H, Chang W-C, Zhang S, Chung S, Yung TJ, Fan JL, Miyamichi K, Luo L, Dan Y. 2016. Cell type-specific long-range connections of basal forebrain circuit. *eLife* 5:1–17. doi:10.7554/*eLife*.13214

Duman RS, Li N, Liu R-J, Duric V, Aghajanian G. 2012. Signaling pathways underlying the rapid antidepressant actions of ketamine. Neuropharmacology 62:35–41. doi:10.1016/J.NEUROPHARM.2011.08.044

Engblom D, Bilbao A, Sanchis-Segura C, Dahan L, Perreau-Lenz S, Balland B, Parkitna JR, Luján R, Halbout B, Mameli M, Parlato R, Sprengel R, Lüscher C, Schütz G, Spanagel R. 2008. Glutamate Receptors on Dopamine Neurons Control the Persistence of Cocaine Seeking. Neuron 59:497–508. doi:10.1016/j.neuron.2008.07.010

Felten A, Montag C, Markett S, Walter NT, Reuter M. 2011. Genetically determined dopamine availability predicts disposition for depression. Brain Behav 1:109–118. doi:10.1002/brb3.20

Friedman AK, Walsh JJ, Juarez B, Ku SM, Chaudhury D, Wang J, Li X, Dietz DM, Pan N, Vialou VF, Neve RL, Yue Z, Han M-H. 2014. Enhancing Depression Mechanisms in Midbrain Dopamine Neurons Achieves Homeostatic Resilience. Science 344:313–319. doi:10.1126/science.1249240

Frisch A, Postilnick D, Rockah R, Michaelovsky E, Postilnick S, Birman E, Laor N, Rauchverger B, Kreinin A, Poyurovsky M, Schneidman M, Modai I, Weizman R. 1999. Association of unipolar major depressive disorder with genes of the serotonergic and dopaminergic pathways. Mol Psychiatry 4:389–392. doi:10.1038/sj.mp.4000536

Fuchikami M, Thomas A, Liu R, Wohleb ES, Land BB, DiLeone RJ, Aghajanian GK, Duman RS. 2015. Optogenetic stimulation of infralimbic PFC reproduces ketamine’s rapid and sustained antidepressant actions. Proc Natl Acad Sci 112:8106–8111. doi:10.1073/pnas.1414728112

Glue P, Gulati A, Le Nedelec M, Duffull S. 2011. Dose- and exposure-response to ketamine in depression. Biol Psychiatry. doi:10.1016/j.biopsych.2010.11.031

Gonçalves L, Sego C, Metzger M. 2012. Differential projections from the lateral habenula to the rostromedial tegmental nucleus and ventral tegmental area in the rat. J Comp Neurol 520:1278–1300. doi:10.1002/cne.22787

Haeffel GJ, Getchell M, Koposov RA, Yrigollen CM, Deyoung CG, Klinteberg BA, Oreland L, Ruchkin V V, Grigorenko EL. 2008. Association between polymorphisms in the dopamine transporter gene and depression: evidence for a gene-environment interaction in a sample of juvenile detainees. Psychol Sci 19:62–9. doi:10.1111/j.1467-9280.2008.02047.x

Hare BD, Pothula S, DiLeone RJ, Duman RS. 2020. Ketamine increases vmPFC activity: Effects of (R)- and (S)-stereoisomers and (2R,6R)-hydroxynorketamine metabolite. Neuropharmacology 166:107947. doi:10.1016/j.neuropharm.2020.107947

Homayoun H, Moghaddam B. 2007. NMDA receptor hypofunction produces opposite effects on prefrontal cortex interneurons and pyramidal neurons. J Neurosci 27:11496–500. doi:10.1523/JNEUROSCI.2213-07.2007

Jendryka M, Palchaudhuri M, Ursu D, van der Veen B, Liss B, Kätzel D, Nissen W, Pekcec A. 2019. Pharmacokinetic and pharmacodynamic actions of clozapine-N-oxide, clozapine, and compound 21 in DREADD-based chemogenetics in mice. Sci Rep 9. doi:10.1038/s41598-019-41088-2

Jönsson EG, Nöthen MM, Grünhage F, Farde L, Nakashima Y, Propping P, Sedvall GC. 1999. Polymorphisms in the dopamine D2 receptor gene and their relationships to striatal dopamine receptor density of healthy volunteers. Mol Psychiatry 4:290–296. doi:10.1038/sj.mp.4000532

Kinoshita H, Nishitani N, Nagai Y, Andoh C, Asaoka N, Kawai H, Shibui N, Nagayasu K, Shirakawa H, Nakagawa T, Kaneko S. 2018. Ketamine-Induced Prefrontal Serotonin Release Is Mediated by Cholinergic Neurons in the Pedunculopontine Tegmental Nucleus. Int J Neuropsychopharmacol 21:305–310. doi:10.1093/ijnp/pyy007

Kumar S, Black SJ, Hultman R, Szabo ST, Demaio KD, Du J, Katz BM, Feng G, Covington HE, Dzirasa K. 2013. Cortical control of affective networks. J Neurosci 33:1116–1129. doi:10.1523/JNEUROSCI.0092-12.2013

Kurdi MS, Theerth KA, Deva RS. 2014. Ketamine: Current applications in anesthesia, pain, and critical care. Anesth essays Res 8:283–90. doi:10.4103/0259-1162.143110

Lammel S, Lim BK, Ran C, Huang KW, Betley MJ, Tye KM, Deisseroth K, Malenka RC. 2012. Input-specific control of reward and aversion in the ventral tegmental area. Nature 491:212–217. doi:10.1038/nature11527

Land BB, Narayanan NS, Liu RJ, Gianessi CA, Brayton CE, M Grimaldi D, Sarhan M, Guarnieri DJ, Deisseroth K, Aghajanian GK, Dileone RJ. 2014. Medial prefrontal D1 dopamine neurons control food intake. Nat Neurosci 17:248–253. doi:10.1038/nn.3625

Laskowski K, Stirling A, McKay WP, Lim HJ. 2011. A systematic review of intravenous ketamine for postoperative analgesia. Can J Anesth. doi:10.1007/s12630-011-9560-0

Lee SJ, Lodder B, Chen Y, Patriarchi T, Tian L, Sabatini BL. 2020. Cell-type-specific asynchronous modulation of PKA by dopamine in learning. Nature 590:451. doi:10.1038/s41586-020-03050-5

Li B, Piriz J, Mirrione M, Chung C, Proulx CD, Schulz D, Henn F, Malinow R. 2011. Synaptic potentiation onto habenula neurons in the learned helplessness model of depression. Nature 470:535–539. doi:10.1038/nature09742

Li N, Lee B, Liu R-J, Banasr M, Dwyer JM, Iwata M, Li X-Y, Aghajanian G, Duman RS. 2010. mTOR-Dependent Synapse Formation Underlies the Rapid Antidepressant Effects of NMDA Antagonists. Science 329:959–964. doi:10.1126/science.1190287

Lindefors N, Barati S, O’Connor WT. 1997. Differential effects of single and repeated ketamine administration on dopamine, serotonin and GABA transmission in rat medial prefrontal cortex. Brain Res 759:205–212. doi:10.1016/S0006-8993(97)00255-2

Liu RJ, Fuchikami M, Dwyer JM, Lepack AE, Duman RS, Aghajanian GK. 2013. GSK-3 inhibition potentiates the synaptogenic and antidepressant-like effects of subthreshold doses of ketamine. Neuropsychopharmacology 38:2268–2277. doi:10.1038/npp.2013.128

Lorrain D., Baccei C., Bristow L., Anderson J., Varney M. 2003. Effects of ketamine and n-methyl-d-aspartate on glutamate and dopamine release in the rat prefrontal cortex: modulation by a group II selective metabotropic glutamate receptor agonist LY379268. Neuroscience 117:697–706. doi:10.1016/S0306-4522(02)00652-8

Manvich DF, Webster KA, Foster SL, Farrell MS, Ritchie JC, Porter JH, Weinshenker D. 2018. The DREADD agonist clozapine N-oxide (CNO) is reverse-metabolized to clozapine and produces clozapine-like interoceptive stimulus effects in rats and mice. Sci Rep 8:3840. doi:10.1038/s41598-018-22116-z

McCullough LD, Sokolowski JD, Salamone JD. 1993. A neurochemical and behavioral investigation of the involvement of nucleus accumbens dopamine in instrumental avoidance. Neuroscience 52:919–925. doi:10.1016/0306-4522(93)90538-Q

Miller OH, Yang L, Wang C-C, Hargroder EA, Zhang Y, Delpire E, Hall BJ. 2014. GluN2B-containing NMDA receptors regulate depression-like behavior and are critical for the rapid antidepressant actions of ketamine. *eLife* 3:e03581. doi:10.7554/*eLife*.03581

Moda-Sava RN, Murdock MH, Parekh PK, Fetcho RN, Huang BS, Huynh TN, Witztum J, Shaver DC, Rosenthal DL, Alway EJ, Lopez K, Meng Y, Nellissen L, Grosenick L, Milner TA, Deisseroth K, Bito H, Kasai H, Liston C. 2019. Sustained rescue of prefrontal circuit dysfunction by antidepressant-induced spine formation. Science 364:eaat8078. doi:10.1126/science.aat8078

Morales M, Margolis EB. 2017. Ventral tegmental area: Cellular heterogeneity, connectivity and behaviour. Nat Rev Neurosci 18:73–85. doi:10.1038/nrn.2016.165

Nair A, Jacob S. 2016. A simple practice guide for dose conversion between animals and human. J Basic Clin Pharm 7:27. doi:10.4103/0976-0105.177703

Oleson EB, Cheer JF. 2013. On the role of subsecond dopamine release in conditioned avoidance. Front Neurosci 7:96. doi:10.3389/fnins.2013.00096

Oleson EB, Gentry RN, Chioma VC, Cheer JF. 2012. Subsecond dopamine release in the nucleus accumbens predicts conditioned punishment and its successful avoidance. J Neurosci 32:14804–14808. doi:10.1523/JNEUROSCI.3087-12.2012

Patriarchi T, Cho JR, Merten K, Howe MW, Marley A, Xiong W-H, Folk RW, Broussard GJ, Liang R, Jang MJ, Zhong H, Dombeck D, von Zastrow M, Nimmerjahn A, Gradinaru V, Williams JT, Tian L. 2018. Ultrafast neuronal imaging of dopamine dynamics with designed genetically encoded sensors. Science 360:eaat4422. doi:10.1126/science.aat4422

Pearson-Fuhrhop KM, Dunn EC, Mortero S, Devan WJ, Falcone GJ, Lee P, Holmes AJ, Hollinshead MO, Roffman JL, Smoller JW, Rosand J, Cramer SC. 2014. Dopamine Genetic Risk Score Predicts Depressive Symptoms in Healthy Adults and Adults with Depression. PLoS One 9:e93772. doi:10.1371/journal.pone.0093772

Peña CJ, Kronman HG, Walker DM, Cates HM, Bagot RC, Purushothaman I, Issler O, Eddie Loh YH, Leong T, Kiraly DD, Goodman E, Neve RL, Shen L, Nestler EJ. 2017. Early life stress confers lifelong stress susceptibility in mice via ventral tegmental area OTX2. Science (80- ) 356:1185–1188. doi:10.1126/science.aan4491

Peña CJ, Smith M, Ramakrishnan A, Cates HM, Bagot RC, Kronman HG, Patel B, Chang AB, Purushothaman I, Dudley J, Morishita H, Shen L, Nestler EJ. 2019. Early life stress alters transcriptomic patterning across reward circuitry in male and female mice. Nat Commun 10:1–13. doi:10.1038/s41467-019-13085-6

Phoumthipphavong V, Barthas F, Hassett S, Kwan AC. 2016. Longitudinal effects of ketamine on dendritic architecture in vivo in the mouse medial frontal cortex. eNeuro 3:91–95. doi:10.1523/ENEURO.0133-15.2016

Picard N, Takesian AE, Fagiolini M, Hensch TK. 2019. NMDA 2A receptors in parvalbumin cells mediate sex-specific rapid ketamine response on cortical activity. Mol Psychiatry 24:828–838. doi:10.1038/s41380-018-0341-9

Pignatelli M, Tejeda HA, Barker DJ, Bontempi L, Wu J, Lopez A, Palma Ribeiro S, Lucantonio F, Parise EM, Torres-Berrio A, Alvarez-Bagnarol Y, Marino RAM, Cai ZL, Xue M, Morales M, Tamminga CA, Nestler EJ, Bonci A. 2020. Cooperative synaptic and intrinsic plasticity in a disynaptic limbic circuit drive stress-induced anhedonia and passive coping in mice. Mol Psychiatry 1–20. doi:10.1038/s41380-020-0686-8

Sarkar A, Kabbaj M. 2016. Sex Differences in Effects of Ketamine on Behavior, Spine Density, and Synaptic Proteins in Socially Isolated Rats. Biol Psychiatry 80:448–456. doi:10.1016/j.biopsych.2015.12.025

Sassano-Higgins S, Baron D, Juarez G, Esmaili N, Gold M. 2016. A REVIEW OF KETAMINE ABUSE AND DIVERSION. Depress Anxiety 33:718–727. doi:10.1002/da.22536

Shinohara R, Taniguchi M, Ehrlich AT, Yokogawa K, Deguchi Y, Cherasse Y, Lazarus M, Urade Y, Ogawa A, Kitaoka S, Sawa A, Narumiya S, Furuyashiki T. 2018. Dopamine D1 receptor subtype mediates acute stress-induced dendritic growth in excitatory neurons of the medial prefrontal cortex and contributes to suppression of stress susceptibility in mice. Mol Psychiatry 23:1717–1730. doi:10.1038/mp.2017.177

Shirayama Y, Hashimoto K. 2017. Effects of a single bilateral infusion of R-ketamine in the rat brain regions of a learned helplessness model of depression. Eur Arch Psychiatry Clin Neurosci 267:177–182. doi:10.1007/s00406-016-0718-1

Sun W, Akins CK, Mattingly AE, Rebec G V. 2005. Ionotropic glutamate receptors in the ventral tegmental area regulate cocaine-seeking behavior in rats. Neuropsychopharmacology 30:2073–2081. doi:10.1038/sj.npp.1300744

Vander Weele CM, Siciliano CA, Matthews GA, Namburi P, Izadmehr EM, Espinel IC, Nieh EH, Schut EHS, Padilla-Coreano N, Burgos-Robles A, Chang C-J, Kimchi EY, Beyeler A, Wichmann R, Wildes CP, Tye KM. 2018. Dopamine enhances signal-to-noise ratio in cortical-brainstem encoding of aversive stimuli. Nature 563:397–401. doi:10.1038/s41586-018-0682-1

Volkow ND, Fowler JS, Gatley SJ, Logan J, Wang G-J, Ding Y-S, Dewey S. 1996. PET Evaluation of the Dopamine System of the Human Brain. J Nucl Med 37.Wahlstrom D, Collins P, White T, Luciana M. 2010. Developmental changes in dopamine neurotransmission in adolescence: Behavioral implications and issues in assessment. Brain Cogn. doi:10.1016/j.bandc.2009.10.013

Wang LP, Li F, Wang Dong, Xie K, Wang Deheng, Shen X, Tsien JZ. 2011. NMDA receptors in dopaminergic neurons are crucial for habit learning. Neuron 72:1055–1066. doi:10.1016/j.neuron.2011.10.019

Wass C, Sauce B, Pizzo A, Matzel LD. 2018. Dopamine D1 receptor density in the mPFC responds to cognitive demands and receptor turnover contributes to general cognitive ability in mice. Sci Rep 8:4533. doi:10.1038/s41598-018-22668-0

Wenzel JM, Oleson EB, Gove WN, Cole AB, Gyawali U, Dantrassy HM, Bluett RJ, Dryanovski DI, Stuber GD, Deisseroth K, Mathur BN, Patel S, Lupica CR, Cheer JF. 2018. Phasic Dopamine Signals in the Nucleus Accumbens that Cause Active Avoidance Require Endocannabinoid Mobilization in the Midbrain. Curr Biol 28:1392-1404.e5. doi:10.1016/j.cub.2018.03.037

Winter S, Dieckmann M, Schwabe K. 2009. Dopamine in the prefrontal cortex regulates rats behavioral flexibility to changing reward value. Behav Brain Res 198:206–213. doi:10.1016/J.BBR.2008.10.040

Wu M, Minkowicz S, Dumrongprechachan V, Hamilton P, Kozorovitskiy Y. 2021. Ketamine rapidly enhances glutamate-evoked dendritic spinogenesis in medial prefrontal cortex through dopaminergic mechanisms. Biol Psychiatry 0. doi:10.1016/j.biopsych.2020.12.022

Yang H, de Jong JW, Tak YE, Peck J, Bateup HS, Lammel S. 2018. Nucleus Accumbens Subnuclei Regulate Motivated Behavior via Direct Inhibition and Disinhibition of VTA Dopamine Subpopulations. Neuron 97:434-449.e4. doi:10.1016/j.neuron.2017.12.022

Yang Y, Cui Y, Sang K, Dong Y, Ni Z, Ma S, Hu H. 2018. Ketamine blocks bursting in the lateral habenula to rapidly relieve depression. Nature 554:317–322. doi:10.1038/nature25509

Zarate CA, Singh JB, Carlson PJ, Brutsche NE, Ameli R, Luckenbaugh DA, Charney DS, Manji HK. 2006. A randomized trial of an N-methyl-D-aspartate antagonist in treatment-resistant major depression. Arch Gen Psychiatry 63:856–864. doi:10.1001/archpsyc.63.8.856

Zweifel LS, Argilli E, Bonci A, Palmiter RD. 2008. Role of NMDA Receptors in Dopamine Neurons for Plasticity and Addictive Behaviors. Neuron 59:486–496. doi:10.1016/j.neuron.2008.05.028